# A component of the *mir-17-92* polycistronic oncomir promotes oncogene-dependent apoptosis

Virginie Olive[1†], Erich Sabio[1†], Margaux J Bennett[1], Caitlin S De Jong[1], Anne Biton[2], James C McGann[1‡], Samantha K Greaney[1], Nicole M Sodir[3], Alicia Y Zhou[4], Asha Balakrishnan[4], Mona Foth[1], Micah A Luftig[5], Andrei Goga[4], Terence P Speed[2], Zhenyu Xuan[6], Gerard I Evan[3], Ying Wan[7], Alex C Minella[8], Lin He[1*]

[1]Department of Molecular and Cell Biology, University of California, Berkeley, Berkeley, United States; [2]Department of Statistics, University of California, Berkeley, Berkeley, United States; [3]Department of Biochemistry, University of Cambridge, Cambridge, United Kingdom; [4]Department of Cell and Tissue Biology, University of California, San Francisco, San Francisco, United States; [5]Department of Molecular Genetics and Microbiology, Duke University, Durham, United States; [6]Department of Molecular and Cell Biology, Center for Systems Biology, University of Texas at Dallas, Dallas, United States; [7]Department of Medicine, The Third Military Medical University, Chongqing, China; [8]Driskill Graduate Program, Department of Medicine, Hematology and Oncology Division, Northwestern University Feinberg School of Medicine, Chicago, United States

*For correspondence: lhe@ berkeley.edu

†These authors contributed equally to this work

‡Present address: Vollum Institute, Oregon Health and Science University, Portland, United States

Competing interests: The authors declare that no competing interests exist.

**Abstract** *mir-17-92*, a potent polycistronic oncomir, encodes six mature miRNAs with complex modes of interactions. In the *Eμ-myc* Burkitt's lymphoma model, *mir-17-92* exhibits potent oncogenic activity by repressing c-Myc-induced apoptosis, primarily through its *miR-19* components. Surprisingly, *mir-17-92* also encodes the *miR-92* component that negatively regulates its oncogenic cooperation with c-Myc. This *miR-92* effect is, at least in part, mediated by its direct repression of Fbw7, which promotes the proteosomal degradation of c-Myc. Thus, overexpressing *miR-92* leads to aberrant c-Myc increase, imposing a strong coupling between excessive proliferation and p53-dependent apoptosis. Interestingly, *miR-92* antagonizes the oncogenic *miR-19* miRNAs; and such functional interaction coordinates proliferation and apoptosis during c-Myc-induced oncogenesis. This *miR-19:miR-92* antagonism is disrupted in B-lymphoma cells that favor a greater increase of *miR-19* over *miR-92*. Altogether, we suggest a new paradigm whereby the unique gene structure of a polycistronic oncomir confers an intricate balance between oncogene and tumor suppressor crosstalk.

## Introduction

MicroRNAs (miRNAs) are a class of small, non-coding RNAs that regulate post-transcriptional gene repression in a variety of developmental and pathological processes (*Ambros, 2004*; *Zamore and Haley, 2005*; *Bartel, 2009*; *Kim et al., 2009*). Due to their small size and the imperfect nature of target recognition, miRNAs have the capacity to regulate many target mRNAs through translational repression and mRNA degradation, thereby acting as global regulators of gene expression (*Lewis et al., 2005*; *Filipowicz et al., 2008*). Unlike mammalian protein-coding genes that follow the one-transcript, one-protein paradigm, many miRNA genes are expressed as polycistronic primary transcripts, generating multiple mature miRNAs under the same transcriptional regulation (*Megraw et al., 2007*). miRNA polycistrons further expand the gene regulatory capacity, since different miRNA components can confer specific yet overlapping biological effects, and their functional interactions can yield unusual complexity.

**eLife digest** The role of genes, in very simple terms, is to be transcribed into messenger RNA molecules, which are then translated into strings of amino acids that fold into proteins. Each of these steps is extremely complex, and a wide range of other molecules can speed up, slow down, stop or otherwise disrupt the expression of genes as protein products. Genes can also code for nucleic acids that are not translated into proteins, such as microRNAs. These are small RNA molecules that can reduce the production of proteins by repressing the translation step and/or by partially degrading the messenger RNA molecules.

*mir-17-92* is a gene that exemplifies much of this complexity. It codes for six different microRNAs in a single primary transcript, and has been implicated in a number of cancers, including lung cancer, Burkitt's lymphoma and other forms of lymphomas and leukemia. One of six microRNAs has a longer evolutionary history than the remaining five: *mir-92* is found in vertebrates, chordates and invertebrates, whereas the other five are only found in vertebrates. However, it is not known how or why the *mir-17-92* gene evolved to code for multiple different microRNAs.

Olive et al. have studied how these *mir-17-92* microRNAs functionally interact in mice with Burkitt's lymphoma, a form of cancer that is associated with a gene called *c-Myc* being over-activated. Mutations in this gene promote the proliferation of cells, and in cooperation with other genetic lesions, this ultimately leads to cancer. *mir-17-92* is implicated in this cancer because it represses the process of programmed cell death (which is induced by the protein c-Myc) that the body employs to stop tumors growing.

Olive et al. found that deleting one of the six microRNAs, *miR-92,* increased the tendency of the *mir-17-92* gene to promote Burkitt's lymphoma. By repressing an enzyme called Fbw7, *miR-92* causes high levels of c-Myc to be produced. While this leads to the uncontrolled proliferation of cells that promotes cancer, it also increases programmed cell death, at least in part, by activating the p53 pathway, a well-known tumor suppression pathway. The experiments also revealed that the action of *miR-92* and that of one of the other microRNAs, *miR-19*, were often opposed to each other. These findings have revealed an unexpected interaction among different components within a single microRNA gene, which acts to maintain an intricate balance between pathways that promote and suppress cancer.

Polycistronic miRNAs often exhibit pleiotropic biological functions with unique gene regulatory mechanisms (*Megraw et al., 2007*). One of the best example is *mir-17-92*, a potent oncomir (i.e., miRNA oncogene), whose genomic amplification and aberrant overexpression have been observed in many human tumors including Burkitt's lymphoma, diffuse large B-cell lymphoma (DLBCL), and lung cancer (*Lu et al., 2005*; *Mendell, 2008*). *mir-17-92* regulates multiple cellular processes during tumor development, including proliferation, survival, angiogenesis, differentiation, and metastasis (*He et al., 2007*; *Uziel et al., 2009*; *Conkrite et al., 2011*; *Nittner et al., 2012*). As a polycistronic oncomir, *mir-17-92* produces a single precursor that yields six individual mature miRNAs (*Figure 1A*, *Figure1—figure supplement 1A*) (*Tanzer and Stadler, 2004*). Based on the seed sequence homology, the six *mir-17-92* components are categorized into four miRNA families (*Figure 1A*, *Figure 1—figure supplement 1A*): miR-17 (miR-17 and 20), miR-18, miR-19 (miR-19a and 19b), and miR-92a (we will designate *miR-92a* as *miR-92* in the remainder of our paper). Interestingly, *miR-92* has a more ancient evolutionary history compared to the other *mir-17-92* components (*Tanzer and Stadler, 2004*). *miR-92* is evolutionarily conserved in vertebrates, chordates, and invertebrates, while the remaining *mir-17-92* components are only found in vertebrates (*Figure 1—figure supplement 1B,C*). Conceivably, the distinct mature miRNA sequence of each *mir-17-92* component determines the specificity of the target regulation. However, the functional significance of the *mir-17-92* polycistronic gene structure remains largely unknown.

The structural analogy to prokaryotic operons has led to the speculation that the co-transcribed *mir-17-92* components can collectively contribute to oncogenesis. However, our studies reveal an unexpected functional interaction among *mir-17-92* components. In the *Eμ-myc* mouse B-cell lymphoma model, while the intact *mir-17-92* acts as an oncogene, its *miR-92* component negatively regulates the oncogenic cooperation with c-Myc. This effect, at least in part, results from the ability of *miR-92* to

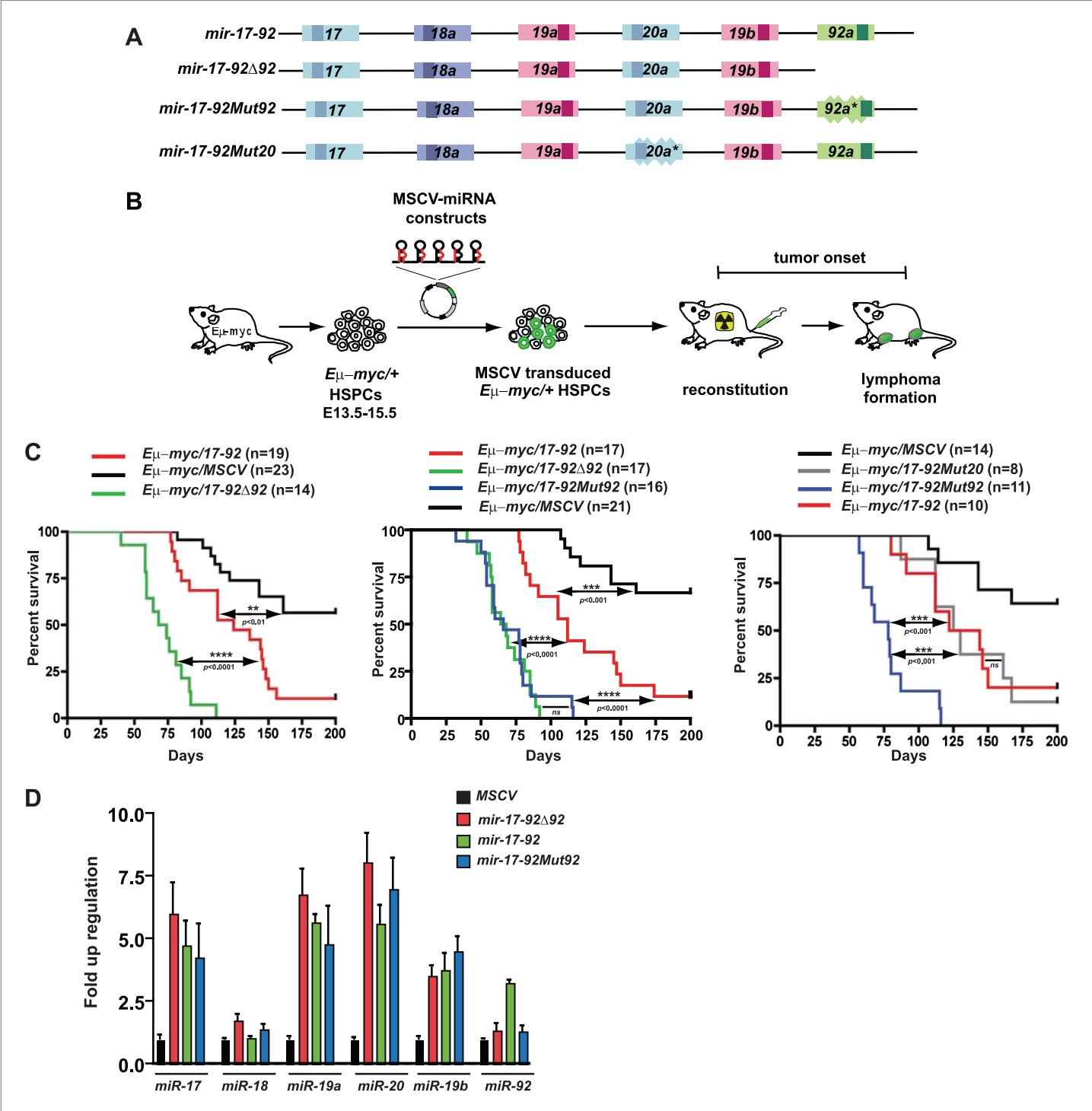

**Figure 1**. *miR-92* negatively regulates the *mir-17-92* oncogenic activity in the *Eµ-myc* B-lymphoma model. (**A**) The gene structure of the *mir-17-92* polycistron and its mutated derivatives. Light colored boxes, pre-miRNAs; dark colored boxes, mature miRNAs. Homologous miRNA components are indicated by the same color. (**B**) Schematic representation of the adoptive transfer protocol using *Eµ-myc* hematopoietic stem and progenitor cells (HSPCs). *Eµ-myc/+* HSPCs were extracted from E13.5–E15.5 mouse embryos, infected with MSCV retroviral vectors overexpressing *mir-17-92* and its derivatives, and finally transplanted into lethally irradiated recipient mice. Lymphoma onset of the adoptive transferred mice was monitored to evaluate the oncogenic collaboration between c-Myc and a specific miRNA. (**C**) *miR-92* deficiency specifically accelerates the oncogenic activity of *mir-17-92* in the *Eµ-myc* model. Using the *Eµ-myc* adoptive transfer model, we compared the oncogenic effects between *mir-17-92* and *mir-17-92Δ92* and observed a significant acceleration of tumor onset in *Eµ-myc/mir-17-92Δ92* mice (p<0.0001, left). When the oncogenic effects of *mir-17-92*,

*Figure 1. Continued on next page*

*Figure 1. Continued*

*mir-17-92Δ92* and *mir-17-92Mut92* were compared in the same adoptive transfer model, *mir-17-92Δ92* and *mir-17-92Mut92* similarly accelerated *Eμ-myc*-induced lymphomagenesis compared to *mir-17-92* (p<0.0001 for both comparisons, middle). Deficiency of *miR-20* failed to affect the oncogenic cooperation between *mir-17-92* and *Eμ-myc*, having minimal effects on tumor onset (right). (**D**) The mutation of *miR-92* has minimal effects on the levels of the remaining *mir-17-92* components. *Eμ-myc* B-lymphoma cells were infected with MSCV retrovirus overexpressing *mir-17-92*, *mir-17-92Δ92* and *mir-17-92Mut92* at an MOI (multiplicity of infection) of 1. Expression levels of *miR-17, 18a, 19a, 20a, 19b* and *92* were subsequently determined using Taqman miRNA assays. Error bars indicate standard deviation (*n* = 3). **p<0.01, ***p<0.001, ****p<0.0001.

The following figure supplements are available for figure 1:

**Figure supplement 1**. Gene structure and evolutionary conservation of *mir-17-92*.

yield aberrant c-Myc dosage, which promotes a strong coupling between oncogene stress and p53-dependent apoptosis. Surprisingly, *miR-92* functionally antagonizes *miR-19*, a key oncogenic *mir-17-92* component, in the context of c-Myc-induced oncogenesis. During B-cell transformation, this *miR-19:miR-92* antagonism is disrupted to favor a greater increase of *miR-19* than *miR-92*. Thus, the polycistronic *mir-17-92* employs an antagonistic interaction among its encoded miRNA components to confer an intricate crosstalk between the oncogene and tumor suppressor networks.

## Results

Since *mir-17-92* is overexpressed in human Burkitt's lymphomas (*Tagawa et al., 2007*), we set out to functionally dissect *mir-17-92* components in the *Eμ-myc* model of Burkitt's lymphoma (*Figure 1B*). The *Eμ-myc* mice carry a *c-myc* transgene downstream of the immunoglobulin (*Ig*) heavy chain enhancer *Eμ* (*Langdon, 1986*; *Adams et al., 1985*), which functionally resembles the *Ig-MYC* translocations that occur frequently in Burkitt's lymphomas (*Tagawa et al., 2007*). The resulting B-cell specific, aberrant c-Myc activation promotes excessive proliferation, yet also evokes potent, p53-dependent apoptosis (*Schmitt et al., 2002*; *Hemann et al., 2003*). Thus, c-Myc-induced apoptosis enables a self-defense mechanism against malignant transformation, producing B-lymphomas with a late onset (*Lowe et al., 2004*). In our adoptive transfer model (*Olive et al., 2009*), *Eμ-myc/+* hematopoietic stem and progenitor cells (HSPCs) were transplanted into lethally irradiated recipient mice, generating chimeric mice that faithfully recapitulated the late tumor onset of the *Eμ-myc* transgenic mice (*Figure 1B*).

When *Eμ-myc/+* HSPCs were infected with MSCV (murine stem cell virus) retrovirus to overexpress the intact *mir-17-92* oncomir, we observed a considerable acceleration in tumor onset compared to the *Eμ-myc*/MSCV control mice (p<0.01, *Figure 1C*). Unexpectedly, the oncogenic cooperation between c-Myc and *mir-17-92* was significantly stronger when *miR-92* was deleted within this oncomir (*Figure 1C*). The average survival of *Eμ-myc/17-92Δ92* mice was 66 days, significantly shorter than that of *Eμ-myc/17-92* mice (112 days, p<0.0001). *mir-17-92Δ92* carried a deletion of *miR-92* pre-miRNA and its flanking sequences, which might alter the expression of the remaining *mir-17-92* components (*Figure 1D*, *Figure 1—figure supplement 1D*). We then engineered a 12-nucleotide *miR-92* seed mutation within *mir-17-92* to abolish the functional *miR-92* with minimal disruption to the overall gene structure. The resulting *mir-17-92Mut92* phenocopied *mir-17-92Δ92* in vivo (*Figure 1C*), significantly enhancing the oncogenic cooperation with c-Myc without altering the level of any remaining *mir-17-92* components (*Figure 1D*, *Figure 1—figure supplement 1D*). This unexpected effect was specifically attributable to *miR-92*. Mutations of *miR-20* or *miR-17* failed to affect oncogenesis in the *Eμ-myc* model (*Figure 1C*, *Figure 1—figure supplement 1D* and data not shown), and mutations of both *miR-19* miRNAs nearly abolished this oncogenic cooperation (*Olive et al., 2009*). This finding suggests that, although *mir-17-92* acted as a potent oncogene as a whole, its *miR-92* component confers an internal negative regulation on its oncogenic cooperation with c-Myc. This effect of *miR-92* clearly contrasts with that of *miR-19*, a key oncogenic *mir-17-92* component that promotes c-Myc-induced lymphomagenesis by repressing apoptosis (*Mu et al., 2009*; *Olive et al., 2009*; *Mavrakis et al., 2010*).

In the *Eμ-myc* model, a strong oncogenic lesion often leads to the B-cell transformation at an earlier developmental stage (*Hemann et al., 2003*). The greater oncogenic activity of *mir-17-92Mut92* in comparison with *mir-17-92* was consistent with *mir-17-92Mut92* preferentially transforming IgM negative progenitor B-cells, and *mir-17-92* frequently transforming IgM positive B-cells (*Figure 2A*; *Table 1*). In comparison to *Eμ-myc/17-92* mice, both *Eμ-myc/17-92Δ92* and *Eμ-myc/17-92Mut92* mice developed more aggressive B-lymphomas, characterized by massive lymph node enlargement, splenic hyperplasia,

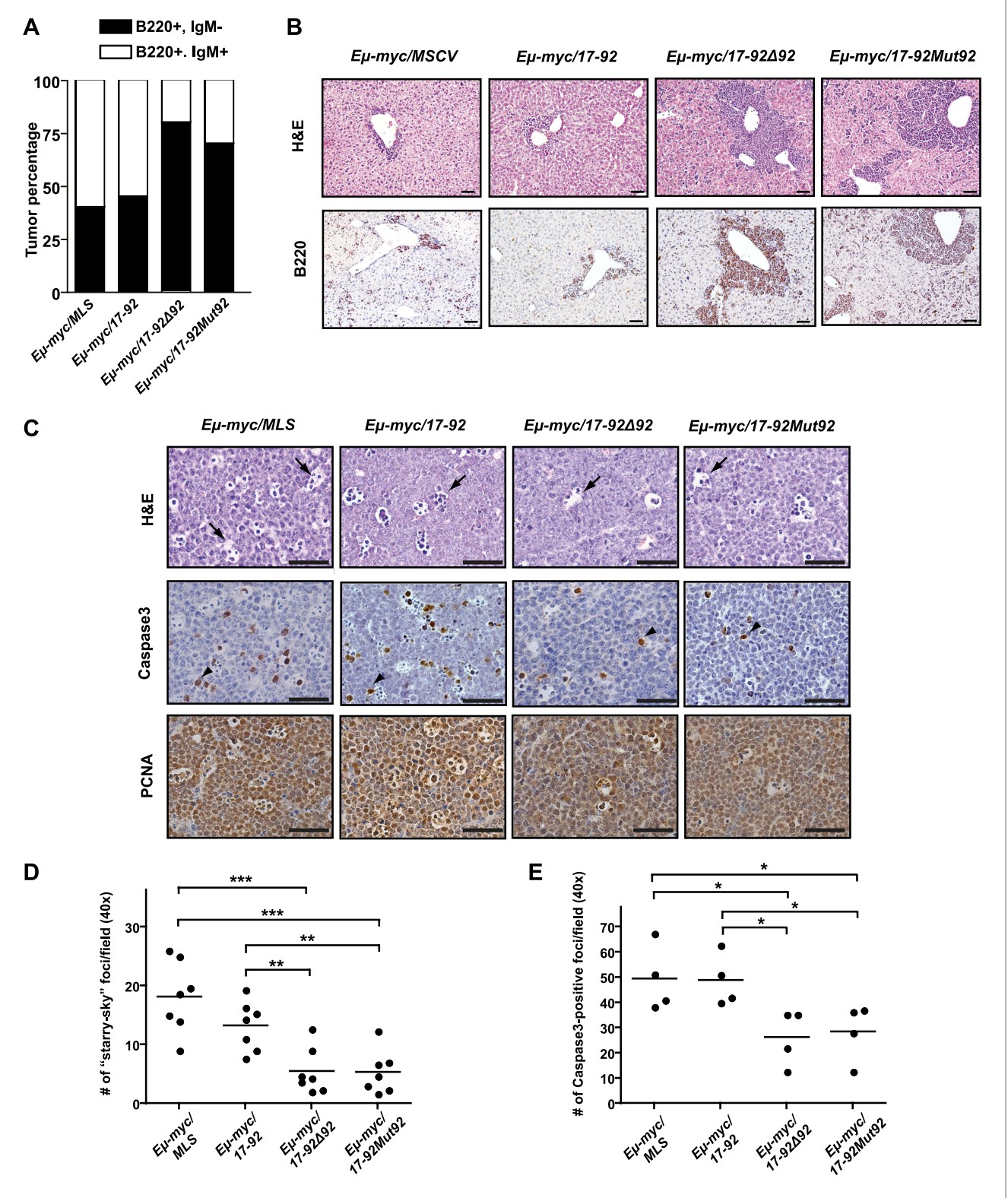

**Figure 2**. The *miR-92* deficient *mir-17-92* cooperates with c-Myc to promote highly aggressive B-lymphomas. (**A**) The percentage of IgM positive and IgM negative B-lymphomas was calculated for each genotype (*Eμ-myc/MSCV*, n = 10; *Eμ-myc/17-92*, n = 9; *Eμ-myc/17-92Δ92*, n = 10; *Eμ-myc/17-92Mut92*, n = 10). (**B**) The *Eμ-myc/17-92Mut92* and *Eμ-myc/17-92Δ92* mice developed high grade B-lymphomas that were frequently disseminated into the liver. When compared to *Eμ-myc/MSCV* and *Eμ-myc/17-92* mice, *Eμ-myc/17-92Mut92* and *Eμ-myc/17-92Δ92* lymphomas gave rise to more liver
*Figure 2. Continued on next page*

*Figure 2. Continued*

dissemination, as indicated by H&E and B220 staining. (**C**) *Eµ-myc/17-92Mut92* and *Eµ-myc/17-92Δ92* lymphomas exhibited a decreased apoptosis compared to *Eµ-myc/MSCV* or *Eµ-myc/17-92* lymphomas. Representative lymphomas were stained for H&E, cleaved caspase-3 and PCNA. Arrow, 'starry sky' feature of apoptotic lymphoma cells; arrowhead, apoptotic cells with positive staining for cleaved caspase-3; scale bar, 50 µm. (**D** and **E**) Apoptosis was quantitatively measured in representative lymphomas of each genotype using the 'starry sky' features (**D**) and cleaved caspase-3 staining (**E**). *p<0.05, **p<0.01, ***p<0.001.

leukemia, and widespread dissemination into visceral organs outside of the lymphoid compartment (*Figure 2B*, data not shown).

During Myc-induced tumorigenesis, aberrant c-Myc dosage yields simultaneous induction of proliferation and apoptosis, imposing a unique selective pressure for pro-survival lesions (*Evan and Vousden, 2001*). Thus, we compared the extent of Myc-induced apoptosis in the *Eµ-myc/17-92*, *Eµ-myc/17-92Δ92*, *Eµ-myc/17-92Mut92*, and control *Eµ-myc/MSCV* lymphomas. The control *Eµ-myc/MSCV* lymphomas invariably exhibited a high proliferation index accompanied by extensive cell death, as evidenced by the widespread 'starry sky' pathology (*Figure 2C,D*) and cleaved caspase 3 staining (*Figure 2C,E*). The potent oncogenic activity of *mir-17-92Δ92* and *mir-17-92Mut92* was consistent with the strong reduction of apoptosis in the lymph node tumors. In comparison, the intact *miR-92* significantly attenuated the repression of c-Myc-induced apoptosis by *mir-17-92* in vivo (*Figure 2C–E*).

We next investigated the effect of *miR-92* alone in regulating c-Myc-induced apoptosis. In the *Eµ-myc* model, *miR-92* overexpression significantly enhanced c-Myc-induced apoptosis in vivo (*Figure 3A,B*, *Figure 3—figure supplement 1A*), consistent with a rapid depletion of *miR-92*-infected cells in premalignant *Eµ-myc* B-cells (*Figure 3—figure supplement 1B*). Similar *miR-92* effects on c-Myc-induced apoptosis were observed in vitro. The *R26^{MER/MER}* mouse embryonic fibroblasts (MEFs) carry a switchable variant of Myc, MycER^{T2}, downstream of the constitutive *Rosa26* promoter, which allows acute activation of the *MycER* transgene by 4-OHT (4-Hydroxytamoxifen) induced nuclear translocation (*Murphy et al., 2008*). The *R26^{MER/MER}* MEFs recapitulate c-Myc-induced apoptosis in vitro, as activated MycER^{T2} induces p53-dependent apoptosis in response to serum starvation (*Murphy et al., 2008*). Enforced *miR-92* expression in *R26^{MER/MER}* MEFs invariably enhanced Myc-induced apoptosis (*Figure 3C*, *Figure 3—figure supplement 1C*).

In addition to promoting c-Myc-induced apoptosis, *miR-92* unexpectedly enhanced c-Myc-induced cell proliferation. A significant increase of BrdU incorporation was observed in *R26^{MER/MER}* MEFs overexpressing *miR-92*, both under normal culture conditions and, more evidently, under serum starvation (*Figure 3D*). The same proliferative effect of *miR-92* was also observed in primary B-cells. Comparison of the proliferative effect of each *mir-17-92* component in bone marrow derived primary

**Table 1.** Flow cytometric immunophenotyping of *Eµ-myc* lymphomas with enforced expression of different *mir-17-92* derivatives

| Genotype | n | Percentage (%) | Immunotype |
|---|---|---|---|
| *Eµ-myc/MSCV* | 4 | 40 | B220+, IgM−, CD19+, CD4−, CD8− |
| | 6 | 60 | B220+, IgM+, CD19+, CD4−, CD8− * |
| *Eµ-myc/17–92* | 4 | 40 | B220+, IgM−, CD19+, CD4−, CD8− |
| | 5 | 50 | B220+, IgM+, CD19+, CD4−, CD8− † |
| | 1 | 10 | B220−, IgM−, CD19−, CD4+, CD8+ |
| *Eµ-myc/17–92Mut92* | 7 | 70 | B220+, IgM−, CD19+, CD4−, CD8− |
| | 3 | 30 | B220+, IgM+, CD19+, CD4−, CD8− ‡ |
| *Eµ-myc/1792Δ92* | 8 | 80 | B220+, IgM−, CD19+, CD4−, CD8− |
| | 2 | 20 | B220+, IgM+, CD19+, CD4−, CD8− § |

*1 out of 6 samples predominantly contains IgM+ cells, with a small percentage of IgM− cells.
†3 out of 5 samples predominantly contain IgM+ cells, with a small percentage of IgM− cells.
‡1 out of 3 samples predominantly contains IgM+ cells, with a small percentage of IgM− cells.
§1 out of 2 samples predominantly contains IgM+ cells, with a small percentage of IgM− cells.

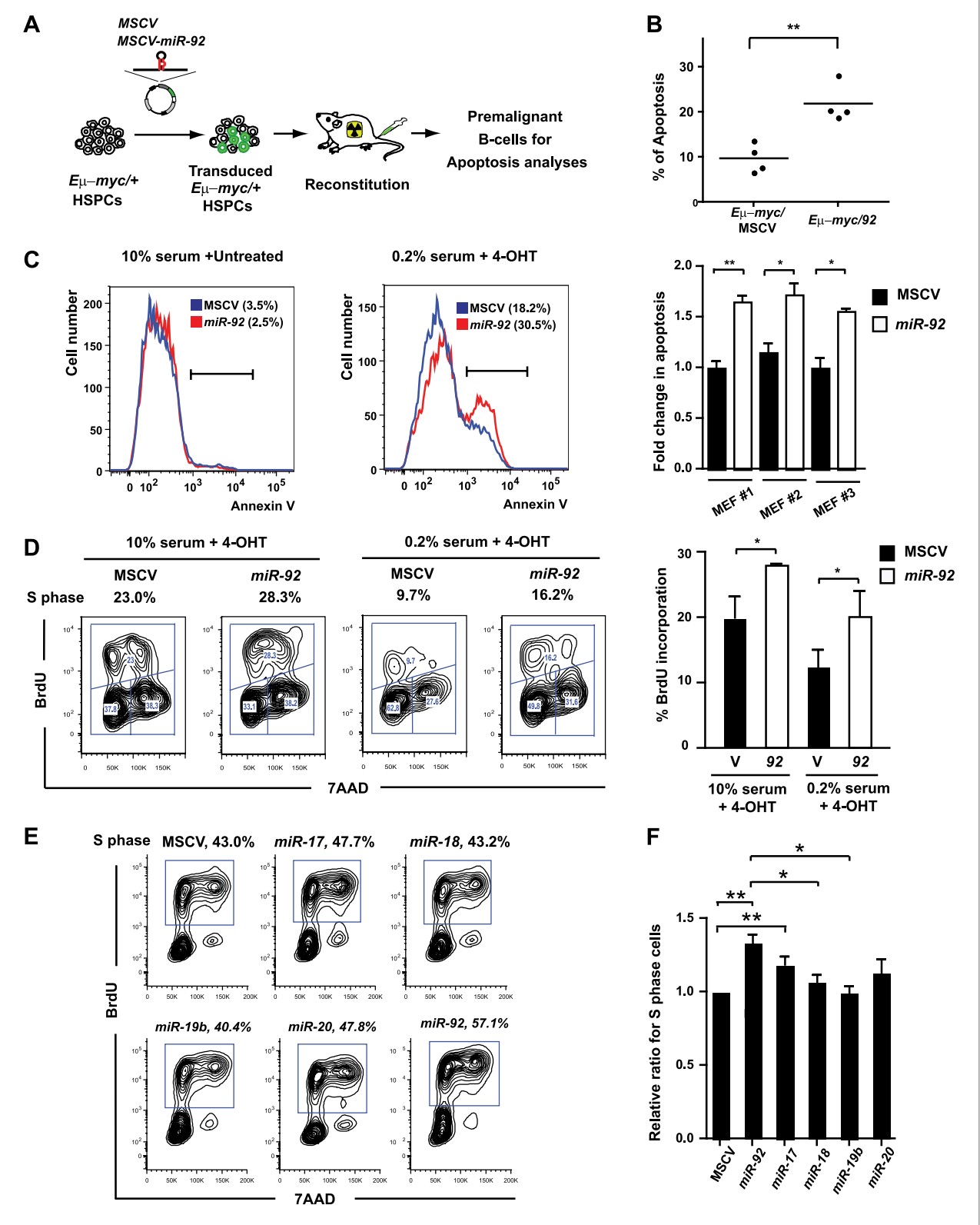

**Figure 3**. *miR-92* enhances both c-Myc-induced apoptosis and c-Myc-induced proliferation. (**A**) The schematic representation of the adoptive transfer model to evaluate the *miR-92* effects on the *Eμ-myc* premalignant B-cells in vivo. (**B**) *miR-92* overexpression enhances the apoptotic response in the premalignant *Eμ-myc* B-cells in vivo. Using the *Eμ-myc* adoptive transfer model, we generated well-controlled *Eμ-myc/MSCV* and *Eμ-myc/92* mice

*Figure 3. Continued on next page*

Figure 3. Continued

reconstituted from donor matched *Eμ-myc* HSPCs. Premalignant *Eμ-myc* splenic B-cells were isolated from the *Eμ-myc/MSCV* and *Eμ-myc/92* mice 6 weeks after reconstitution. The in vivo apoptosis was measured by the level of caspase activation using Red-VAD-FMK, a fluorescently labeled caspase inhibitor that specifically bound to cleaved caspases. The percentage of *Eμ-myc* B-cells positive for cleaved caspases was shown for four independent experiments. (**C**) Enforced *miR-92* expression in *R26^MER/MER* MEFs significantly enhanced c-Myc-induced apoptosis. *miR-92* overexpressing and the control *R26^MER/MER* MEFs were serum starved, and the MycER^T2 transgene was activated by 4-OHT treatment. The level of apoptosis of each MEF was measured using Annexin V staining before (left) and after (middle) 4-OHT treatment and serum starvation. Quantification of c-Myc-induced apoptosis was performed in three independent MEF lines that overexpressed MSCV or *miR-92* (right panel, error bars represent SEM). (**D**) Enforced *miR-92* expression in *R26^MER/MER* MEFs significantly enhanced c-Myc-induced proliferation. Proliferative effects of *miR-92* was measured by BrdU incorporation in MycER^T2 activated *R26^MER/MER* MEFs. *miR-92* cooperated with c-Myc to promote BrdU incorporation in both 10% (left) and 0.2% (middle) serum culture conditions. Quantification of BrdU incorporation was performed in two independent experiments (right). (**E**) *miR-92* is a potent *mir-17-92* component to promote primary B-cell proliferation. The proliferative effects of all *mir-17-92* miRNAs were measured individually in primary B-cells using BrdU incorporation. (**F**) The quantification of BrdU incorporation in experiments described in (**E**) was performed in four independent experiments. Error bars represent standard deviation, *p<0.05, **p<0.01.

The following figure supplements are available for figure 3:

**Figure supplement 1**. *miR-92* enhances c-Myc-induced apoptosis both in vitro and in vivo.

B-cells revealed that the *miR-92* component yielded one of the strongest effects (**Figure 3E,F**). In addition, *miR-92* deficiency significantly compromised the ability of *mir-17-92* to promote cell cycle progression in B-cells (**Figure 3—figure supplement 1D**). Interestingly, strong proliferative effects have been reported for nearly all *mir-17-92* components, yet the exact cell type and biological context can select specific components as the predominant drivers for cell proliferation. Taken together, our data suggest that *miR-92* is a unique *mir-17-92* component that functionally couples c-Myc-induced cell proliferation and c-Myc-induced apoptosis in the B-cell compartment.

To investigate the molecular mechanism underlying *miR-92* functions, we performed microarray analyses comparing gene expression profiles of *R26^MER/MER* MEFs overexpressing *miR-92* or the control MSCV vector. These MEFs were serum starved and 4-OHT treated to trigger strong Myc-induced apoptosis. *miR-92*-upregulated genes were significantly enriched for the cell cycle pathway, including *ccnd1*, *ccnb1*, *ccnb2*, *cdc25b*, *cdc25c,* and *cdk4* (**Figure 4A,B**), consistent with the ability of *miR-92* to promote Myc-induced cell proliferation. Genes upregulated by *miR-92* were also enriched for the *p53* pathway, including the classic *p53* target *mdm2*, as well as the pro-apoptotic p53 targets—*noxa*, *bax*, *puma*, *perp*, and *bid* (**Figure 4A,B**, **Figure 4—figure supplement 1A**). Since aberrant c-Myc activation triggered a p53-dependent apoptotic response (**Lowe et al., 2004**), our observation is consistent with *miR-92* further enhancing p53 activation downstream of c-Myc. Interestingly, *p21*, a canonical p53 target, was not induced by *miR-92* in the MycER^T2 activated *R26^MER/MER* MEFs (**Figure 4—figure supplement 1A**). It is likely that the transcriptional repression of *p21* by c-Myc renders *p21* irresponsive to p53 activation under this biological context (**Heasley et al., 2002**). Using real-time PCR, we validated the ability of *miR-92* to induce cell cycle genes and activate p53 targets in both *R26^MER/MER* MEFs, as well as primary B-cells (**Figure 4C**, **Figure 4—figure supplement 1A,B**). Hence, the molecular signature imposed by *miR-92* overexpression is consistent with its functional readout.

The activation of the p53 pathway by c-Myc is essential for the induction of the apoptotic response in the *Eμ-myc* model (**Schmitt et al., 2002**). A major mechanism that governs Myc-induced p53 activation is the transcriptional induction of the gene encoding *Arf*, which inhibits Mdm2-mediated p53 ubiquitination and degradation (**Lowe et al., 2004**; **Campaner and Amati, 2012**). The ability of *miR-92* to enhance c-Myc-induced apoptosis and to increase the expression of p53 targets raised the possibility that *miR-92* overexpression activates p53 possibly through elevated Arf. In both *R26^MER/MER* MEFs and wild-type primary B-cells, *miR-92* overexpression alone caused significant accumulation of Arf mRNA and protein (**Figure 4C,D**, **Figure 4—figure supplement 1C**), consistent with the rapid stabilization of the p53 protein (**Figure 4D**, **Figure 4—figure supplement 1C**) without alteration of *p53* mRNA (**Figure 4—figure supplement 1D**). Notably, the ability of *miR-92* to induce p53 activation occurred not only in 4-OHT treated *R26^MER/MER* MEFs with MycER^T2 activation, but also in untreated *R26^MER/MER* MEFs with normal c-Myc level. This was clearly demonstrated by the elevation of p53 protein level, as well as the increased p53 target expression (**Figure 4—figure supplement 1B,C**).

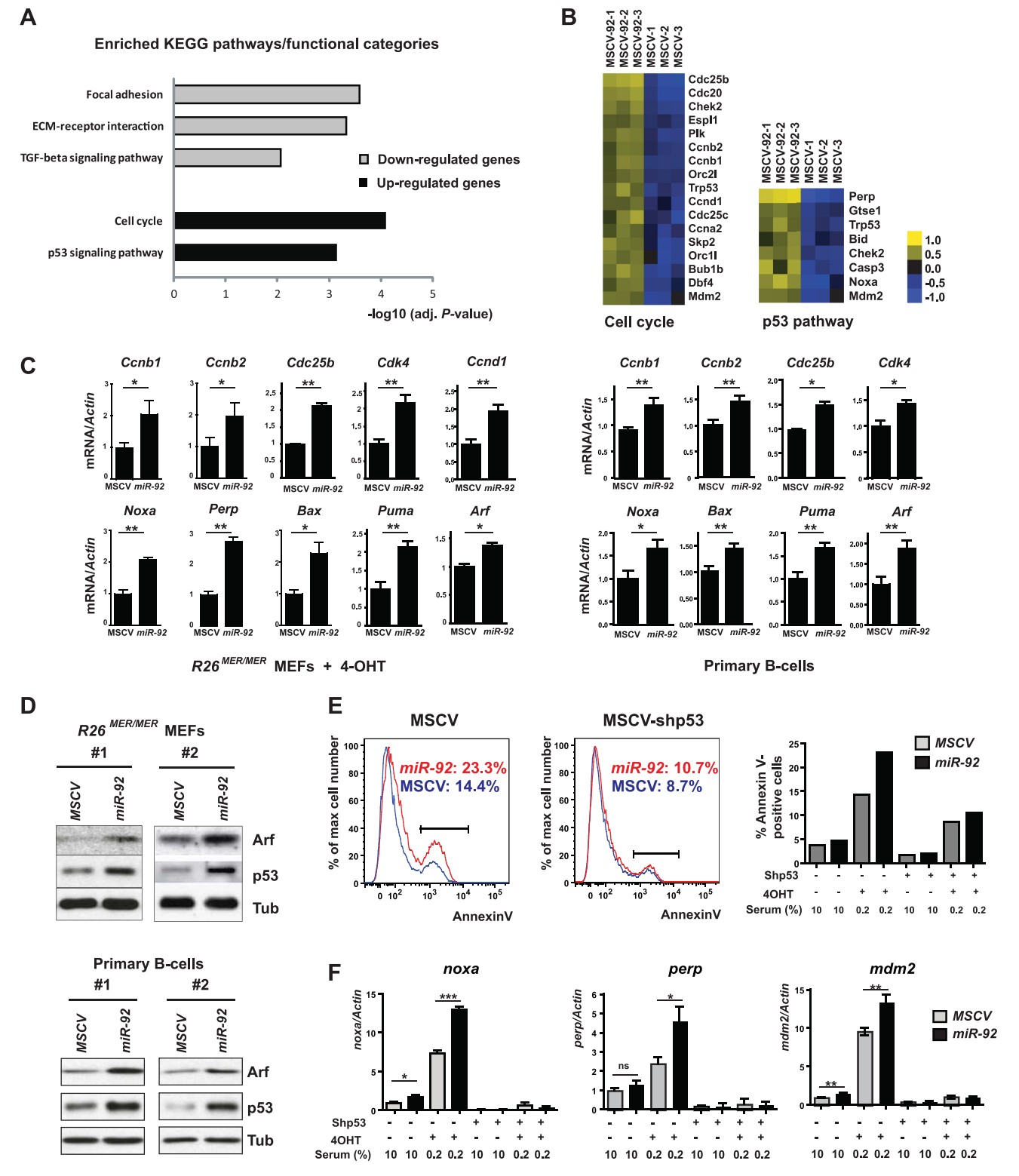

**Figure 4**. *miR-92* induces apoptosis through the activation of the p53 pathway. (**A**) The genes upregulated by *miR-92* were enriched for the cell cycle pathway and the *p53* pathway. Microarray analyses compared gene expression profiles of serum starved and 4-OHT treated *R26^MER/MER* MEFs overexpressing either *miR-92* or a control MSCV vector (n = 3). The differentially expressed genes were defined as those with at least 1.5-fold expression level change using SAM (Significance analysis of microarrays, false discovery rate <1%). Pathway analyses were performed on upregulated and downregulated genes
*Figure 4. Continued on next page*

*Figure 4. Continued*

using the KEGG database. (**B**) The heatmaps of the *miR-92* upregulated genes enriched for the cell cycle and p53 pathways. (**C**) Components of the cell cycle and p53 pathways were upregulated upon *miR-92* overexpression in both MEFs (left) and primary B-cells (right). The quantitation of gene expression was performed using real time PCR. (**D**) *miR-92* overexpression induces the accumulation of Arf and p53 proteins in MEFs and primary B-cells from bone marrow. Western analyses were performed on the *R26^MER/MER* MEFs (left) and primary B-cells (right) that overexpressed *miR-92* or a control MSCV vector in two independent experiments. The infected *R26^MER/MER* MEFs were assayed at 6 hr after serum starvation and 4-OHT treatment; the infected primary B-cells were collected 72 hr post infection. (**E**) The apoptotic effect of *miR-92* requires an intact p53 pathway. We infected *R26^MER/MER* MEFs with two MSCV retrovirus, MSCV-p53shRNA and MSCV-92, to obtain doubly infected cells. Knocking down *p53* in *R26^MER/MER* MEFs abolished the ability of *miR-92* to enhance c-Myc-induced apoptosis, as measured by Annexin V staining (two left panels). The percentage of apoptotic MEFs of each experimental condition was quantitatively measured (right). (**F**) The induction of the p53 pathway components by *miR-92* is dependent on an intact p53. Knocking down *p53* in *R26^MER/MER* MEFs abolished the ability of *miR-92* to induce pro-apoptotic p53 targets and other canonical p53 targets, including *noxa, perp* and *mdm2*. Error bars represent standard deviation, *p<0.05, **p<0.01, ***p<0.001.
The following figure supplements are available for figure 4:

**Figure supplement 1**. *miR-92* overexpression triggers the activation of the p53 pathway.

The induction of p53 by *miR-92* prompted us to investigate the functional importance of p53 in *miR-92*-induced apoptotic response. Knockdown of *p53* in *R26^MER/MER* MEFs not only led to a suppression of c-Myc-induced apoptosis, but also completely abolished the effect of *miR-92* to enhance c-Myc-induced apoptosis (*Figure 4E*). These findings suggested that an intact p53 pathway is required for the apoptotic effect of *miR-92*. Consistently, the *miR-92* induction of the pro-apoptotic genes, including *noxa, perp,* and *mdm2*, also was mediated by the intact p53 (*Figure 4F*). Thus, aberrant c-Myc activation triggers an apoptotic response through p53 activation; and co-expression of *miR-92* with c-Myc leads to an even stronger p53 activation, and subsequently apoptotic response.

Our findings suggest parallels between *c-myc* and *miR-92*: both are potent oncogenes that promote excessive cell proliferation coupled with p53-dependent apoptosis, and both are capable to induce expression of cell cycle genes (*ccnb1, ccnd1, cdk4,* and *cdc25*) (*Lowe et al., 2004*; *Campaner and Amati, 2012*) and p53 pathway components (*Arf, puma, noxa, perp,* and *mdm2*) (*Lowe et al., 2004*; *Campaner and Amati, 2012*). The functional analogy between c-Myc and *miR-92*, as well as the molecular overlap between their downstream pathways, led us to investigate the effect of *miR-92* on c-Myc. Intriguingly, *miR-92* expression significantly enhanced c-Myc protein level both in MEFs and in primary B-cells (*Figure 5A*), without affecting the *c-myc* mRNA level (*Figure 5—figure supplement 1A*, data not shown). Consistent with the stabilization of endogenous c-Myc, *miR-92* overexpression in *R26^MER/MER* MEFs stabilized the MycER^T2 protein (*Figure 5B*). The dosage of c-Myc protein is crucial for its biological readout (*Murphy et al., 2008*). While c-Myc dosage determines the extent of cell cycle gene induction and cell proliferation, it also regulates the degree of p53 activation and subsequent apoptosis (*Murphy et al., 2008*) (*Figure 5—figure supplement 1B*). Thus, the ability of *miR-92* to induce aberrant c-Myc accumulation likely constitutes the molecular basis for its ability to promote both cell proliferation and p53-dependent apoptosis.

Based on our findings, we speculated that *miR-92* targets could include negative regulators of c-Myc protein accumulation. Therefore, we searched genes known to negatively regulate *c-myc* for the presence of putative *miR-92* binding sites. Using the Targetscan and RNA22 algorithms (*Lewis et al., 2005*; *Miranda et al., 2006*; *Bartel, 2009*), we identified eight candidate *miR-92* targets, each of which contained one or more predicted *miR-92* binding sites in the 3′ untranslated region (3′UTR). Real-time PCR analysis of these candidate genes confirmed *fbw7* (F-box and WD repeat domain-containing 7) as a likely target of *miR-92* (*Figure 5C*, *Figure 5—figure supplement 1C*). *fbw7*, which contains two *miR-92* target sites within its 3′UTR (*Figure 5C*), is the substrate recognition component of an SCF-type E3 ubiquitin ligase that mediates the degradation of several proto-oncoproteins, including Myc, Cyclin E, c-Jun, and Notch (*Welcker and Clurman, 2008*; *Crusio et al., 2010*; *Wang et al., 2012*). A luciferase reporter or a FLAG-tagged *fbw7*-encoding ORF (open reading frame), when fused to the wild-type *fbw7* 3′ UTR, were both significantly repressed in a *miR-92* dependent manner (*Figure 5D*, *Figure 5—figure supplement 1D*). Yet enforced *miR-92* expression failed to repress the luciferase reporter that contained an *fbw7* 3′UTR with two mutated *miR-92* binding sites (*Figure 5D*), suggesting that *miR-92* binding to *fbw7* 3′UTR is required for this repression. Furthermore, *miR-92* effectively repressed endogenous Fbw7 protein level, as demonstrated by the decreased *fbw7* mRNA level and

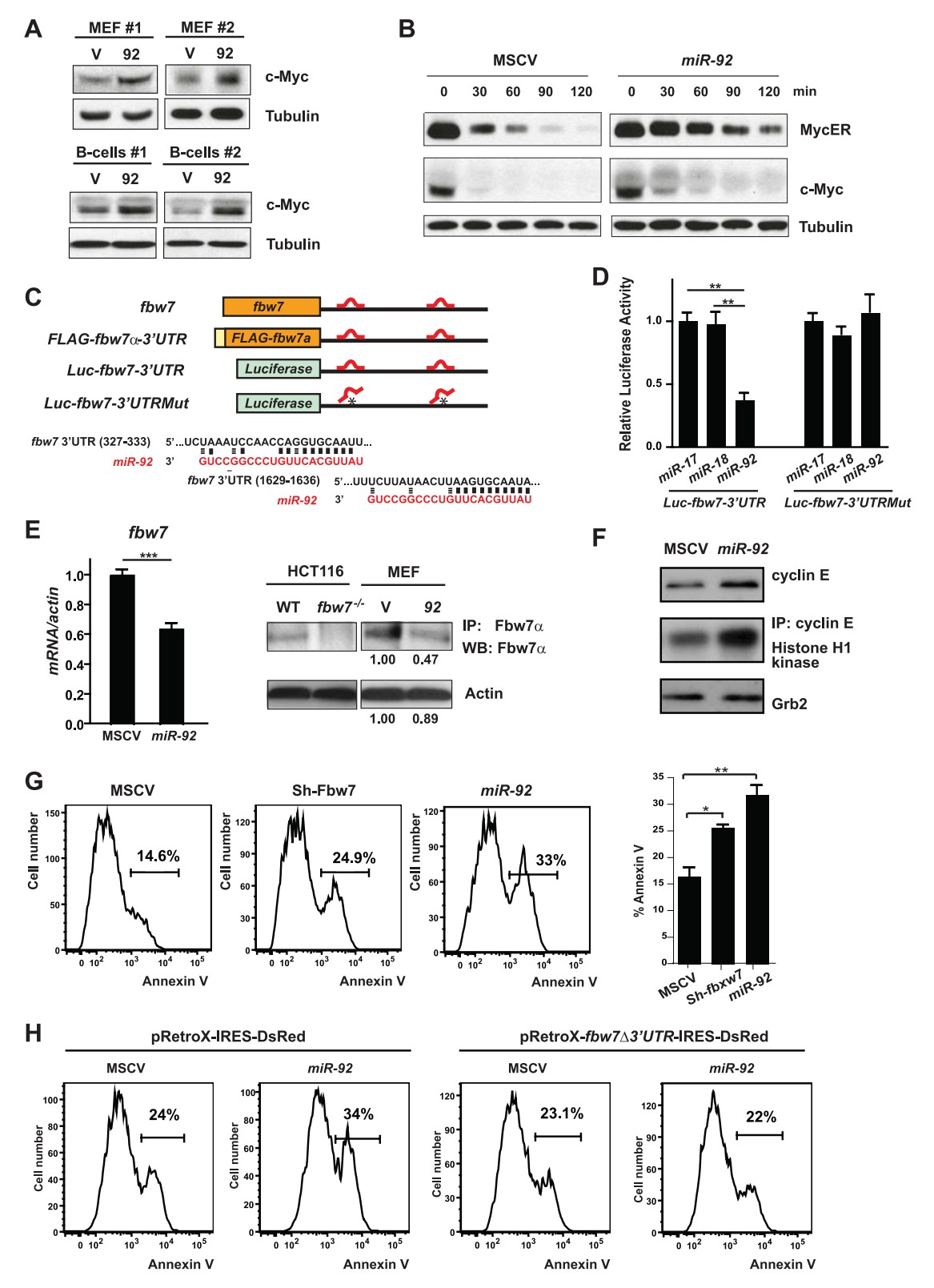

**Figure 5**. *miR-92* promotes the accumulation of c-Myc protein through repressing Fbw7. (**A**) *miR-92* enhances the accumulation of c-Myc protein in synchronized *R26^MER/MER* MEFs (upper), as well as primary B-cells (lower). The *miR-92* overexpression and the control *R26^MER/MER* MEFs were synchronized by serum starvation and were collected 12 hr after being released into serum culture conditions to determine the c-Myc protein level. This synchronization

*Figure 5. Continued on next page*

*Figure 5. Continued*

approach in *R26^MER/MER* MEFs has provided us with the most consistent measurement for c-Myc protein level, because it is regulated in a cell-cycle-dependent manner. (**B**) *miR-92* overexpression decreases the turnover of c-Myc protein. Serum-synchronized *R26^MER/MER* MEFs that overexpress either *miR-92* or the control MSCV vector were released into the serum for 6 hr, treated with cycloheximide, collected at the indicated time points, then analyzed by western blot to determine the levels of MycER and the endogenous c-Myc protein. (**C**) Schematic representation of the two *miR-92* binding sites in the murine *fbw7* 3'UTR. Additionally, a luciferase reporter and a *FLAG* tagged *fbw7* ORF were each placed upstream of a wild-type *fbw7* 3'UTR, or a mutated *fbw7* 3'UTR that abolished the predicted *miR-92* binding. (**D**) The expression of *Luc-fbw7-3'UTR* was specifically repressed by *miR-92* in *Dicer^−/−* HCT116, while mutations of the two putative *miR-92* binding sites within the *fbw7-3'UTR (Luc-fbw7-*3'UTRMut) abolished this repression. (**E**) The endogenous *fbw7* gene was downregulated by *miR-92* post-transcriptionally. Both the endogenous *fbw7* mRNA (left) and the endogenous Fbw7 protein (right) were repressed upon *miR-92* overexpression in *R26^MER/MER* MEFs. Due to the lack of a proper antibody to detect endogenous Fbw7 in regular western analysis, we demonstrated the downregulation of endogenous Fbw7 by *miR-92* using immunoprecipitation followed by immunoblotting with a polyclonal anti-Fbw7 antibody. (**F**) *miR-92* enhances the accumulation of Cyclin E protein. Overexpression of *miR-92* increased the accumulation of Cyclin E protein, which was further confirmed by the increased Cyclin E-dependent kinase activity. (**G**) The knockdown of *fbw7* resembles the effect of *miR-92* to enhance c-Myc-induced apoptosis. Knocking down *fbw7* in *R26^MER/MER* MEFs enhanced c-Myc-induced apoptosis, partially recapitulating the phenotype caused by *miR-92* overexpression. Apoptosis was quantitatively measured by Annexin V staining in two independent lines of *R26^MER/MER* MEFs upon serum starvation and 4-OHT treatment. (**H**) Overexpression of *fbw7* abolished the apoptotic effects of *miR-92* in *R26^MER/MER* MEFs. *R26^MER/MER* MEFs were doubly infected by pRetro-*fbw7αΔ3'UTR*-IRES-dsRed and MSCV-*miR-92*. The c-Myc-induced apoptosis was quantitatively measured by Annexin V staining in doubly infected *R26^MER/MER* MEFs upon serum starvation and 4-OHT treatment. Error bars represent standard deviation). *p<0.05; **p<0.01.

The following figure supplements are available for figure 5:

**Figure supplement 1**. *miR-92* overexpression enhances c-Myc protein level by repressing Fbw7.

Fbw7 immunoprecipitation (***Figure 5E***). Consistent with *fbw7* as an important target for *miR-92*, enforced *miR-92* expression upregulated multiple Fbw7 substrates at their protein levels, including c-Myc and cyclinE (***Figure 5F***). Observations from our in vivo experiments also supported this post-transcriptional regulation of Fbw7 by *miR-92*, as we observed an inverse correlation between the level of *miR-92* and *fbw7* when comparing *Eμ-myc/17-92Δ92 and Eμ-myc/17-92* lymphoma cells (***Figure 5—figure supplement 1E***).

*fbw7* has previously been postulated as a potential *miR-92* target based on the presence of *miR-92* target sites (***Mavrakis et al., 2011***), yet it remains unclear how *fbw7* mediated the pro-apoptotic effects of *miR-92*, given its well-characterized functions as a tumor suppressor. Recent findings indicate that the acute inactivation of tumor suppressor Fbw7 imposes a strong oncogenic stress to induce p53-dependent apoptosis, conferring a selective advantage to cells with deficient p53 function (***Minella et al., 2007***; ***Onoyama et al., 2007***; ***Matsuoka et al., 2008***; ***Grim et al., 2012***). This p53-dependent apoptosis is, at least in part, due to an aberrant increase of c-Myc dosage (***Onoyama et al., 2007***; ***Matsuoka et al., 2008***). These findings suggested that a major mechanism through which *miR-92* enhanced the c-Myc protein level, and subsequently, c-Myc-induced apoptosis, could be through its direct repression of Fbw7. In support of this hypothesis, *miR-92* overexpression significantly increased the c-Myc protein level in wild-type Hct116 cells, but not in *FBW7^−/−* Hct116 cells (***Figure 5—figure supplement 1F***), suggesting FBW7 was essential for *miR-92* to induce c-MYC increase. Functionally, acute *fbw7* knockdown in *R26^MER/MER* MEFs partially phenocopied the effect of *miR-92* to enhance c-Myc-induced apoptosis (***Figure 5G***, ***Figure 5—figure supplement 1G***); while overexpression of an *fbw7α* open reading frame (ORF), albeit above its physiological level, completely abolished this apoptotic effect of *miR-92* (***Figure 5H***, ***Figure 5—figure supplement 1H***). Nevertheless, it is still likely that additional mechanisms downstream of *miR-92* also promote its apoptotic effects, because *fbw7* knockdown largely recapitulated the extent of c-Myc upregulation by *miR-92* (***Figure 5—figure supplement 1G***), yet only partially phenocopied its pro-apoptotic effects (***Figure 5G***). In addition, overexpression of *fbw7* above its physiological level might amplify the extent of functional interactions between *fbw7* and *miR-92* in regulating apoptosis. Despite these caveats, our results strongly argue that the *miR-92*-Fbw7 axis constitutes a major mechanism underlying the pro-apoptotic effects of *miR-92*.

Downregulation of Fbw7 by *miR-92* significantly enhanced the protein level of c-Myc in *R26^MER/MER* MEFs and in primary B-cells (***Figure 5A***). It is conceivable that the ability of *miR-92* to repress Fbw7 in vivo could similarly enhance the c-Myc accumulation in the *Eμ-myc/92* premalignant B-cells, promoting rapid cell proliferation and a p53-dependent apoptotic response. Unfortunately, due to technical limitations,

we were not able to demonstrate an increased c-Myc protein level as a result of *miR-92* overexpression in the *Eμ-myc* premalignant B-cells. There was a significant depletion of the *Eμ-myc/92* premalignant B-cells due to excessive apoptosis (*Figure 3—figure supplement 1B*), making it difficult to collect enough cells to analyze the c-Myc protein level by western analyses. Similarly, we could not obtain enough cells to compare the protein level of c-Myc in the premalignant B-cells from the *Eμ-myc/17-92*, *Eμ-myc/17-92Δ92,* and *Eu-myc/MSCV* animals. Nevertheless, our functional studies in cell culture, combined with the inverse expression correlation between *fbw7* and *miR-92* in vivo, strongly argue the importance of the *miR-92*-Fbw7-Myc axis to promote the pro-apoptotic effects of *miR-92*.

In the context of the c-Myc cooperation, *mir-17-92* encodes miRNA components with opposing biological functions. While *miR-19* miRNAs repress c-Myc-induced apoptosis to promote *Eμ-myc* lymphomagenesis (*Mu et al., 2009*; *Olive et al., 2009*), *miR-92* enhances c-Myc-induced apoptosis to attenuate the tumorigenic effects. Consistent with the opposing effects of *miR-19* and *miR-92*, co-expression of these two miRNAs as a dicistron attenuated the apoptotic effect of *miR-92* in premalignant *Eμ-myc* B-*cells* in vivo (*Figure 6A,B*, *Figure 6—figure supplement 1A*). A similar antagonistic interaction was also observed in *R26^{MER/MER}* MEFs; and introducing a *miR-19b* mutation within *mir-19b-92* dicistron abolished this interaction (*Figure 6C*). Since *miR-19* represses *pten* to promote the PI3K/AKT pathway, the activation of AKT signaling would lead to increased phosphorylation of Mdm2, thus destabilizing p53 to dampen the apoptotic response induced by *miR-92* (*Gottlieb et al., 2002*; *Ogawara et al., 2002*). Consistent with this hypothesis, we observed a decreased p53 induction and an unaltered c-Myc level when *miR-92* was co-expressed with *miR-19* (*Figure 6D*, *Figure 6—figure supplement 1B*).

This *miR-19:miR-92* antagonism appears to be conserved evolutionarily. In *Xenopus laevis*, *miR-19* and *miR-92* have identical sequence to their mammalian orthologs (*Figure 1—figure supplement 1B*). Based on Targetscan and RNA22 miRNA target prediction algorisms (*Lewis et al., 2003*, *2005*; *Miranda et al., 2006*; *Grimson et al., 2007*), their target specificity is also conserved for key miRNA targets, although the exact binding sites may or may not be conserved (*Olive et al., 2009*) (*Figure 6—figure supplement 1C*). In addition, the biological functions of *miR-19* and *miR-92* exhibit evolutionary conservation between *Xenopus laevis* and mammals. Individual injection of *miR-19* promoted cell survival of hydroxyurea-treated *Xenopus* embryos, while co-injection of *miR-19a* and *miR-92* significantly attenuated this pro-survival effect (*Figure 6E*). This functional antagonism was specific for *miR-92* and *miR-19*, since co-injection of other *mir-17-92* components or a mutated *miR-92* did not yield any functional interactions in combination with *miR-19* (*Figure 6F*).

Given the opposing biological effects of *miR-19* and *miR-92* during c-Myc-induced lymphoma development, differential regulation of these two miRNA families could determine the oncogenic activity of *mir-17-92*. Under normal physiological conditions, this *miR-19:miR-92* antagonism could attenuate the detrimental oncogenic signaling by inducing apoptosis in cells with inappropriate *mir-17-92* induction. During malignant transformation, and particularly during c-Myc-induced oncogenesis, this *miR-19:miR-92* antagonism could be disrupted to favor cell survival. Using real time PCR analyses, we compared the relative abundance of *miR-19*a, *miR-19*b, and *miR-92* in normal splenic B-cells, premalignant *Eμ-myc* B-cells, and *Eμ-myc* lymphomas (*Figure 7A*). Comparing to normal splenic B-cells, the levels of all three mature miRNA species were elevated in both premalignant and malignant *Eμ-myc* B-cells, possibly due to transcriptional activation of *mir-17-92* by c-Myc (*Donnell et al., 2005*). However, the *miR-19* to *miR-92* ratios significantly increased during c-Myc-induced lymphomagenesis (*Figure 7B*). In other words, when normalized to the respective miRNA levels in normal splenic B-cells, mature *miR-19* (including *miR-19a* and *miR-19b*) exhibited a greater increase in premalignant and malignant *Eμ-myc* B-cells than mature *miR-92* (*Figure 7A–C*). This differential increase was most evident in premalignant *Eμ-myc* B-cells; the fully transformed *Eμ-myc* B-lymphoma cells exhibited a lesser difference (*Figure 7A,B*). This observation is consistent with premalignant *Eμ-myc* B-cells having an intact p53-dependent apoptotic response, thus a stronger selective pressure for a greater *miR-19:miR-92* ratio. In comparison, most *Eμ-myc* B-lymphomas have a defective p53 response, hence a less strong selective pressure to maintain a high *miR-19:miR-92* ratio. We also validated this observation using northern analysis. Comparing normal splenic B-cells and multiple *Eμ-myc* lymphoma cells, the levels of the mature *miR-19a, miR-19b* and *miR-92* were all elevated in transformed B-cells; however, the degree of increase for *miR-19a* and *miR-19b* was significantly higher than that of *miR-92* (*Figure 7C*). This differential increase of *miR-19* and *miR-92* was also observed in human Burkitt's lymphoma cell lines when compared to normal B-cells isolated from the periphery blood (*Figure 7D*). More importantly, this phenomenon was not limited to

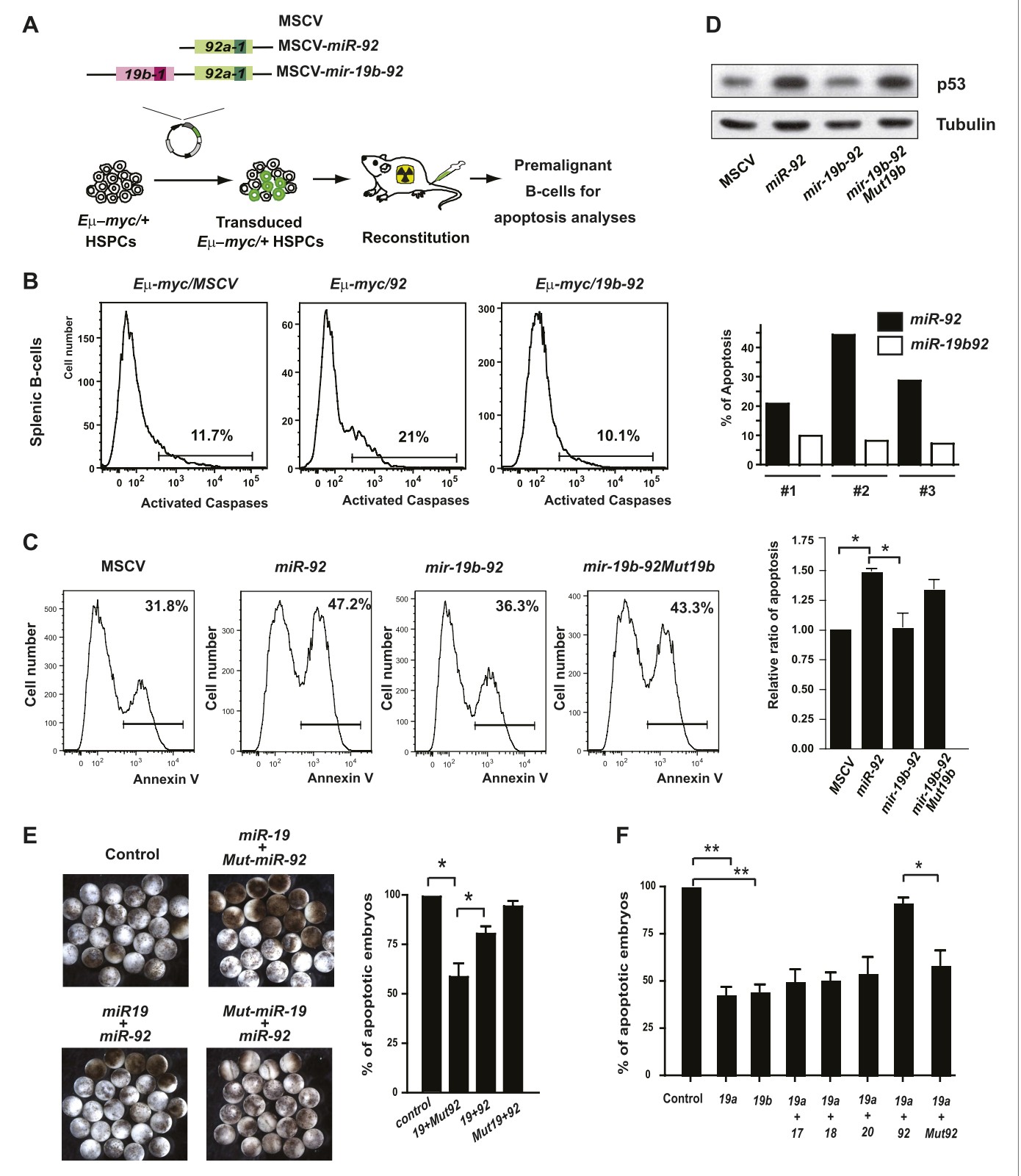

**Figure 6**. The antagonistic interaction between *miR-19* and *miR-92* regulates the balance between proliferation and apoptosis. (**A**) The schematic representation of the *Eμ-myc* adoptive transfer model to evaluate the functional interaction between *miR-92* and *miR-19* in vivo. Light colored boxes, pre-miRNAs; dark colored boxes, mature miRNAs. (**B**) *miR-19* antagonizes the apoptotic effects of *miR-92* in vivo. *miR-92* overexpression in the *Eμ-myc*

*Figure 6. Continued on next page*

*Figure 6. Continued*

adoptive transfer model enhanced apoptosis in premalignant *Eµ-myc* splenic B-cells, while the *mir-19b-92* dicistron expression abolished this apoptotic effect (left three panels). A quantitative analysis of apoptosis by FACS was shown for three independent, well-controlled experiments (right). (**C**) *miR-19b* dampens the *miR-92*-induced apoptosis in MycER$^{T2}$ activated *R26$^{MER/MER}$* MEFs. *R26$^{MER/MER}$* MEFs were infected by *miR-92*, *mir-19b-92*, *mir-19b-92Mut19b* and the MSCV control vector, and were subsequently serum starved and treated with 4-OHT to activate MycER$^{T2}$. Apoptosis in these samples was measured quantitatively using Annexin V staining (left four panels). The extent of apoptosis induced by MSCV, *miR-92*, *mir-19b-92*, *mir-19b-92Mut19b* was normalized to that of MSCV infected *R26$^{MER/MER}$* MEFs and then averaged from four independent experiments (right). (**D**) *miR-19b* dampens the *miR-92*-induced p53 activation. *R26$^{MER/MER}$* MEFs that overexpress the indicated constructs (*miR-92*, *mir-19b-92Mut19b* and *mir-19b-92*) were collected 48 hr after infection and then analyzed by western blot to determine the level of p53 protein. (**E**) *miR-92* and *miR-19* exhibit antagonistic effects to regulate hydroxyurea (HU)-induced cell death in *Xenopus* embryos. Representative images of HU-treated *Xenopus* embryos that were co-injected with human Ago2 and the indicated miRNA mimics (left). Co-injection of *miR-92* dampened the cell survival effects of *miR-19* on HU-induced apoptosis (right, n = 3, with >20 embryos in each group). (**F**) *miR-92* exhibits a specific antagonistic interaction with *miR-19*. Injection of *miR-19a* or *miR-19b* rescued HU-induced apoptosis in *Xenopus* embryos. Co-injection of *miR-92*, but not a mutated *miR-92*, or other *mir-17-92* components, dampened the cell survival effect of *miR-19* (n = 3, with >20 embryos in each group). Error bars represent standard deviation, *p<0.05; **p<0.01.

The following figure supplements are available for figure 6:

**Figure supplement 1**. Functional antagonism between *miR-19:miR-92* regulates the balance between proliferation and apoptosis.

---

*c-myc* driven B-lymphomas. In the LT2-MYC murine model of hepatocellular carcinoma (HCC), where tumor development was initiated by tetracycline-inducible c-Myc expression, *miR-19a* and *miR-19b* also exhibited a stronger increase than *miR-92* when comparing tumor cells and the normal counterpart (*Figure 7E*).

These observations were consistent with a previous finding, where the inducible *c-myc* activation in a human Burkitt's lymphoma cell line induced both *miR-19a* and *miR-19b* to a greater extent than *miR-92* (*Donnell et al., 2005*). Although *miR-19* and *miR-92* are co-transcribed from the *mir-17-92* precursor, the differential increase of *miR-19* vs *miR-92* occurs in multiple c-Myc-driven tumor types. Thus, the relative abundance of *miR-19* and *miR-92* could constitute an important molecular basis to regulate the initiation and progression of c-Myc-induced tumor development.

## Discussion

The unique polycistronic structure of *mir-17-92* constitutes the basis for its pleiotropic functions and the complex mode of interactions among its miRNA components. A high level of *mir-17-92* in normal or premalignant cells could lead to suboptimal consequences that are counter-balanced through an intrinsic negative regulation by *miR-92* (*Figure 7F*). As we demonstrated in vitro where *miR-92*, by directly downregulating Fbw7, enhances c-Myc protein level to promote apoptosis, the ability of *miR-92* to repress Fbw7 in vivo could similarly constitute a major mechanism to enhance c-Myc-induced apoptosis. This effect of *miR-92* is a double edged sword in c-Myc driven tumors, as its overexpression gives rise to a strong and obligated coupling between excessive proliferation and a potent, p53-dependent apoptosis (*Figure 7F*). This coupling is consistent with the previous observation that a lower level of constitutive c-Myc acts more effectively to promote tumor initiation, while a higher level of c-Myc is selected by the terminal tumors with defective apoptosis machinery (*Murphy et al., 2008*). Therefore, *mir-17-92* encodes an internal component to confer a negative regulatory feedback on its oncogenic activity, imposing a strong selection for anti-apoptotic lesions to shape the path of malignant transformation. More interestingly, c-Myc transcriptionally activates *mir-17-92* that encodes *miR-92* (*Hemann et al., 2005*), which in turn enhances c-Myc dosage, at least in part, by repression Fbw7. It is possible that aberrant c-Myc activation triggers a positive feedback loop to further increase c-Myc dosage to strengthen the apoptotic response and to eliminate cells with oncogenic potential. It is worth noting that the *miR-92* apoptotic effect described in this study depends on an intact p53 response. Consequently, in terminal *Eµ-myc* B-lymphoma cells that often carry a defective p53 response, *miR-92* failed to enhance c-Myc-induced apoptosis (*Mu et al., 2009*).

The functional readout of *miR-92* heavily depends on cell types and biological contexts. It is important to recognize that *miR-92* is not a tumor suppressor miRNA. Like c-Myc, *miR-92* elicits potent oncogene stress to engage tumor suppressor response, at least in part, by activating p53. In the premalignant *Eµ-myc/92* B-cells, the effect of *miR-92* to repress Fbw7 most likely results in an increase of c-Myc level, which coupled with the intact p53 response to strongly sensitize the cells to *miR-92*-induced apoptosis.

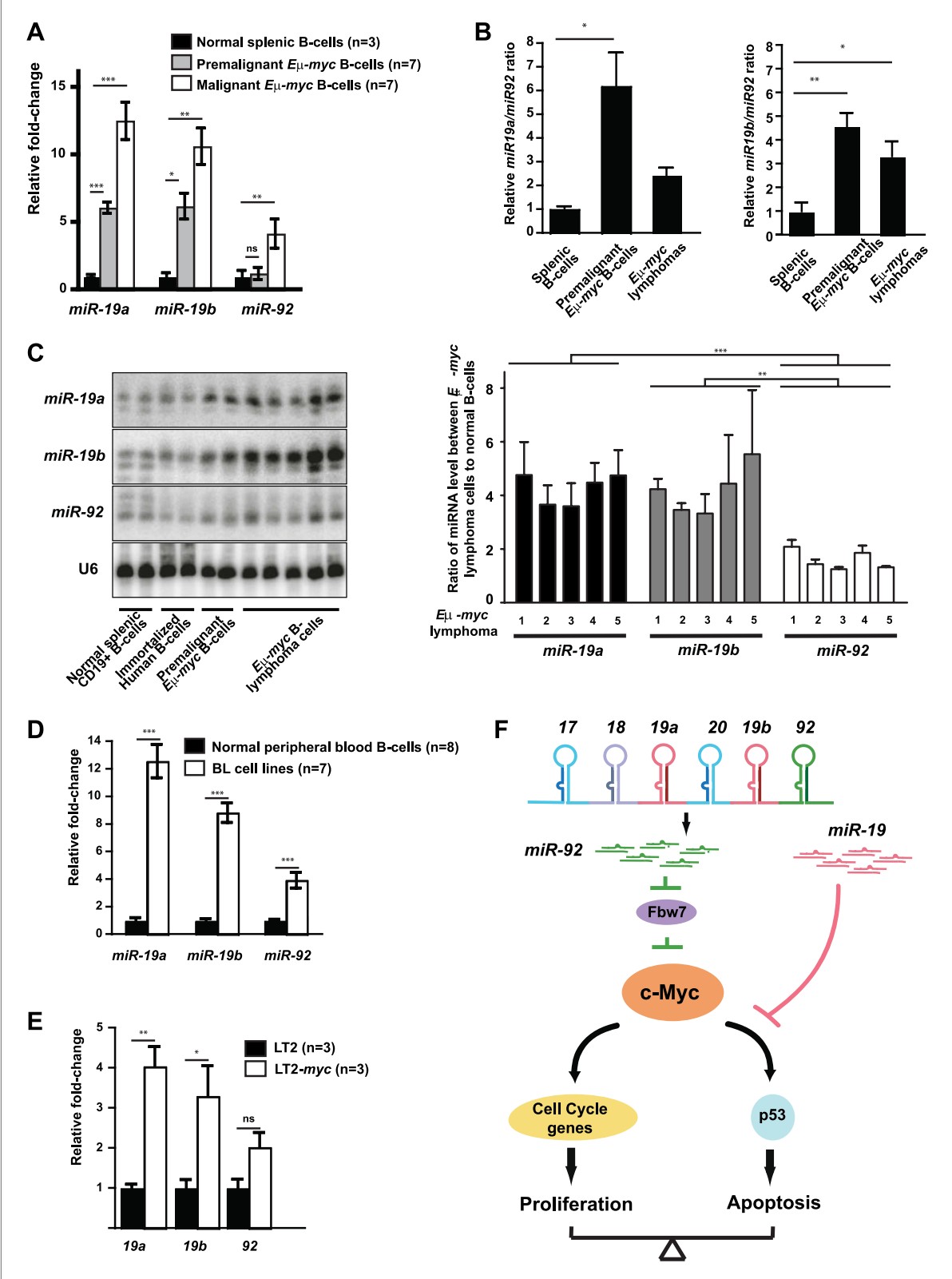

**Figure 7**. The *miR-19:miR-92* antagonism is disrupted during malignant transformation. (**A** and **B**) Compared to normal splenic B-cells, premalignant and malignant *Eμ-myc* B-cells favored a greater increase in mature *miR-19* (*miR-19a* and *miR-19b*) than *miR-92*. The purified normal splenic B-cells, premalignant *Eμ-myc* bone marrow B-cells and malignant *Eμ-myc* B-lymphoma cells were subjected to Taqman miRNA assays to determine the expression level of
*Figure 7. Continued on next page*

*Figure 7. Continued*

*miR-19a, miR-19b* and *miR-92.* Comparing premalignant/malignant *Eμ-myc* B-cells vs normal splenic B-cells, all three miRNAs exhibited an increased level, although the increase in *miR-19a* or *miR-19b* was significantly higher than that of *miR-92* (**A**). In the same experiment, the relative ratios for *miR-19a:miR-92* and *miR-19b:miR-92* were measured for all normal splenic B-cells and *Eμ-myc* B-cells (**B**). (**C**) Mature *miR-19* and *miR-92* are differentially expressed in normal splenic B-cells and *Eμ-myc* B-lymphoma cells. The normal splenic B-cells, immortalized human B-cells, premalignant *Eμ-myc/+* B-cells, and *Eμ-myc/+* B-lymphoma cells were subjected to Northern analysis. Compared to normal splenic B-cells, both malignant and premalignant *Eμ-myc/+* B-cells favored a greater increase of *miR-19* than *miR-92*. (**D**) Compared to normal B-cells isolated from peripheral blood, human Burkitt's lymphoma cell lines favor a greater increase in mature *miR-19* than *miR-92*. (**E**) Compared to normal livers (LT2), mouse hepatocellular carcinomas caused by the inducible c-Myc over-expression (LT2-*myc*) favor a greater increase in mature *miR-19* than *miR-92*. (**F**) A diagram describes our proposed model to explain the functional interactions between *miR-92* and *miR-19* in c-Myc-induced B-lymphomagenesis. Aberrant c-Myc expression couples rapid proliferation and p53-dependent apoptosis. *miR-92* overexpression further increases c-Myc dosage to strengthen this coupling, at least in part by repressing Fbw7. This *miR-92* effect ensures a potent mechanism to eliminate premalignant c-Myc overexpressing cells. Interestingly, *miR-92* and can be antagonized by the survival effects of the *miR-19* miRNAs encoded by the same *mir-17-92* miRNA polycistron. Taken together, while *miR-19* miRNAs repressed c-Myc-induced apoptosis to promote the oncogenic cooperation between *mir-17-92* and c-Myc, *miR-92* exhibits a negative regulation. Thus, the antagonistic interactions between *miR-92* and *miR-19* confer an intricate crosstalk between proliferation and apoptosis. Error bars represent standard deviation, *p<0.05; **p<0.01, ***p<0.001.

Under other contexts when proliferation becomes a rate-limiting event for oncogenesis, or when p53-dependent apoptosis is compromised, *miR-92* could render a pro-proliferative effect that is strictly oncogenic (*Tsuchida et al., 2011*). Likewise, the functional readout of other *mir-17-92* components also heavily depends on cell types and biological contexts. *miR-19* promotes c-Myc-induced B-lymphomas by repressing apoptosis (*Mu et al., 2009*; *Olive et al., 2009*), yet has little effects in promoting *Rb*-deficient retinoblastomas (*Conkrite et al., 2011*); *miR-17* allows the bypass of Ras-induced senescence by promoting proliferation (*Hong et al., 2010*), yet fails to affect c-Myc-induced lymphomas, possibly due to its functional redundancy with c-Myc.

Both cooperative and antagonistic interactions operate among subsets of *mir-17-92* components. The *miR-19:miR-92* antagonism constitutes a novel mechanism to confer an intricate balance between oncogene signaling and innate tumor suppressor responses (*Figure 7F*). This balance can be disrupted in premalignant and malignant cells that exhibit c-Myc overexpression, as an increase in the *miR-19:miR-92* ratio is likely to favor the suppression of c-Myc-induced apoptosis and to promote oncogenesis. Although all *mir-17-92* components are co-transcriptionally regulated, different changes of *miR-19* vs *miR-92* during oncogenesis could be a result of differential miRNA biogenesis and/or turn-over. It has been shown that specific RNA-binding proteins, such as hnRNP A1, promote the processing of a specific *mir-17-92* component, *miR-18* (*Guil and Cáceres, 2007*). Future studies are likely to reveal important mechanisms underlying cell type- and context-dependent differential regulation of *mir-17-92* components, which will generate important insights on the biology of polycistronic miRNAs.

Our current study mostly focuses on the antagonistic interaction between *miR-19* and *miR-92* in c-Myc driven oncogenesis, yet it reveals a more general mechanism underlying the structural function relationship of polycistronic miRNAs. It is likely that the complex interactions among polycistronic miRNA components can coordinate and balance a multitude of cellular and molecular processes during normal development and disease. Interestingly, in the case of *mir-17-92*, *miR-92* has a different evolutionary history compared to the other *mir-17-92* components. *miR-92* is evolutionary conserved in Deuterostome (including vertebrates and chordates), Ecdysozoa (including flies and worms), and Lophotrochozoa, yet the remaining *mir-17-92* components are only found in vertebrates (*Figure 1—figure supplement 1C*). The functional antagonism between the more ancient *miR-92* and the newly evolved *mir-19* might result from the convergence of these two separate evolutionary paths at the origin of vertebrates. This antagonism could evolve to regulate cell proliferation and cell death downstream or independent of c-Myc in both normal development and disease. Thus, our studies suggest a novel mechanism by which a crosstalk between oncogene and tumor suppressor pathways has been hardwired through evolution into the unique gene structure of a polycistronic oncomir.

## Materials and methods

### Molecular cloning

*mir-17-92 Δ92* and *mir-17-92* were amplified by PCR and subsequently cloned into the XhoI and EcoRI sites of the MSCV retrovirus vectors. In these vectors, miRNAs were placed downstream of the LTR

promoter, which is followed either by a SV40-GFP cassette (for all in vivo experiments), a PGK-Puro-IRES-GFP cassette, or a SV40-CD4 cassette (for in vitro experiments) (*Hemann et al., 2005*). To construct MSCV-*17-92Mut92*, MSCV-*17-92Mut20*, and MSCV-*17-92Mut19b* vectors, a 12-nucleotide mutation was introduced into the seed region of the mature *miR-92*, *miR-20*, or *miR-19b* using the Quikchange XL mutagenesis kit (200521; Stratagene) and the following primers:

*Mut20* primers: GACAGCTTCTGTAGCACTAAtaaacaataatcGCAGGTAGTGTTTAGTTATC and GATAACTAAACACTACCTGCGATTATTGTTTATTAGTGCTACAGAAGCTGTC.

*Mut92* primers: CAATGCTGTGTTTCTGTATGGTtaacattaacatCCGGCCTGTTGAGTTTG and CAAACTCAACAGGCCGGATGTTAATGTTAACCATACAGAAACACAGCATTG.

*Mut19b* primers: CTGTGTGATATTCTGCTGacatttaagtacCAAAACTGACTGTGGTAGTG and CACTACCACAGTCAGTTTTGGTACTTAAATGTCAGCAGAATATCACACAG.

The loss of *miR-92, miR-20* or *miR-19b* expression and the intact expression level of the remaining *mir-17-92* components were validated using the TaqMan MicroRNA Assays (4427975; Applied Biosystems, Foster City, CA). *mir-19b-92, mir-19bMut92*, and *mir-19b-92Mut19b* were similarly amplified by PCR (ACTGCTCGAGAGCTTCGGCCTGTCGCCC and GTAGAATTCATGTATCTTGTAC) from the *mir-17-92, mir-17-92Mut92*, and *mir-17-92Mut19b* construct described above and subsequently cloned into the XhoI and EcoRI sites of the MSCV retrovirus vectors.

To construct the MSCV-Shp53 vector, shRNA against p53 was placed downstream of the LTR promoter of the MSCV-SV40-HuCD4 retroviral vector (*Xue et al., 2007*). MSCV-Shfbw7 construct was kindly provided by Dr Hans Guido Wendel (*Mavrakis et al., 2011*). To construct the pRetroX-fbw7-IRES-DsRedExpress (*Xu et al., 2010*), *fbw7α* ORF was placed downstream of the LTR promoter followed by an IRES-DsRed cassette.

## Adoptive transfer of *Eµ-myc* HSPCs for lymphomagenesis

The hematopoietic stem and progenitor cells (HSPCs) were isolated from E13.5-E15.5 *Eµ-myc/+* mouse embryos and were transduced with MSCV alone or MSCV vectors expressing various *mir-17-92* derivatives. The MSCV retroviral vector used in our adoptive transfer model contains a SV40-GFP cassette that allows us to monitor transduced HSPCs both in vitro and in vivo. Infected HSPCs were subsequently transplanted into an 8- to 10-week-old, lethally irradiated C57BL/6 recipient mice. Tumor onset was subsequently monitored by weekly palpation, and tumor samples were either collected into formalin for histopathological studies, or prepared as single cell suspension for FACS analysis and for cell culture studies. Both the *Eµ-myc/+* mice and the recipient mice are on C57BL/6 background.

## LT2-MYC mouse liver tumor model

The LT2-MYC mouse model for human hepatocellular carcinoma (HCC) is a double transgenic mouse model, in which the tetracycline transactivator protein (tTA) is driven by the hepatocyte-specific promoter, the liver activator protein (LAP) promoter, while the human c-MYC gene is driven by the tetracycline response element (TRE). The LT2-MYC model exhibits 'dox-off' regulation, where c-Myc expression is turned on in hepatocytes in the absence of doxycycline.

LT2-MYC mice taken off doxycycline-containing food, between 3–5 weeks of age, develop distinct tumor nodules around 8–12 weeks on an average (*Kistner et al., 1996*; *Shachaf et al., 2004*). Total RNA was extracted from liver tumor samples from three independent mice, as well as normal livers from the doxycycline treated LT2 mice. Total RNAs were prepared using Trizol (15596018; Invitrogen) and subjected to real time PCR analyses as described below.

## Cell culture and retroviral infection

Primary murine B-cells were prepared from bone marrows of 4- to 6-week-old mice and were cultured in RPMI with 10% fetal bovine serum (FBS), 50 µM beta-mercaptoethanol (M3148; Sigma) and 2 ng/ml Il-7 (407-ML-005; R&D). *R26^MER/MER* and *R26^MER/+* MEFs were kindly provided by Gerald Evan's laboratory. MEFs were cultured in DMEM with 10% fetal bovine serum. *Eµ-myc* tumor cells were derived from lymphomas from the terminal-stage *Eµ-myc* animals. *Eµ-myc* lymphoma cells overexpressing various *mir-17-92* derivatives were cultured in 45% DMEM, 45% IMDM with 10% fetal bovine serum, and 50 µM β-mercaptoethanol (M3148; Sigma) on irradiated NIH-3T3 feeder cells. Immortalized human B-cell lines were cultured in RPMI with 10% FBS and 90 µM beta-mercaptoethanol. *Dicer*-deficient Hct116 cells, kindly provided by Dr Bert Vogelstein (*Cummins et al., 2006*), and *Fbxw7*-deficient Hct116 cells (*Grim et al., 2012*) were cultured in McCoy's 5A media with 10% fetal bovine

serum. Human Burkitt's lymphoma cell lines, including BL41, BL2, MutuI, Daudi, Raji (provided by Dr Terry Rabbitts), Manca, and Jiyoje were cultured in RPMI with 10% FBS.

Mouse primary B-cell cultures or MEFs were infected by MSCV retroviruses expressing various *mir-17-92* derived miRNA clusters, shRNA against p53 (*Xue et al., 2007*), shRNA against fbw7 (*Mavrakis et al., 2011*), or fbw7 cDNA (pRetroX-fbw7-IRES-DsRedExpress). In *Figure 4E,F*, double infection was performed to obtain *R26^{MER/MER}* MEFs that co-expressed shRNA *p53* and *miR-92*. In this experiment, MEFs were initially infected with an ecotropic MSCV-*p53shRNA-SV40huCD4* retrovirus to a nearly 100% infection efficiency, as validated by FACS analysis using huCD4 antibody. The second infection was achieved using an amphotropic MSCV-*miR-92-PGK-Puro-IRES-GFP* retrovirus. Doubly infected cells were then selected using puromycin. In *Figure 5H*, double infection of *R26^{MER/MER}* MEFs with pRetroX-*fbw7*-IRES-DsRedExpress and MSCV-*miR-92-PGK-Puro-IRES-GFP* were similarly performed. For all experiments with primary murine B-cells, bone marrow cells were cultured for 48 hr before retroviral infection and collected or analyzed 72 hr after infection. After 5 days in culture, the percentage of B220-positive cell is 100%. In *Figure 3E*, *Figure 3—figure supplement 1*, B-cells were infected with MSCV retrovirus containing a PGK-Puro-IRES-GFP cassette. FACS analysis was performed after gating on the GFP-positive population. In *Figure 5A*, the collected B-cells were infected with retrovirus containing SV40-CD4 cassette. Infected cells were purified with Human CD4 Microbeads (130-045-101; Miltenyi Biotec) using MACS Purification Columns MS (130-042-201; Miltenyi Biotec).

## The collection of normal and malignant B-cells in vivo

Normal mouse B-cells were isolated from the spleen or the bone marrow of 4- to 6-week-old C57B/6J mice, using CD19 Micro-Beads (Miltenyi Biotec) or by negative selection (Easysep 19754; STEMCELL). Similarly, premalignant *Eμ-myc* B-cells were extracted from the bone marrow of 5- to 6-week-old *Eμ-myc* transgenic mice. Malignant *Eμ-myc* B-cells were extracted from the lymph node tumors of terminal-stage *Eμ-myc* mice. In addition, the normal human B-cells from peripheral blood were FACS sorted from the peripheral blood of healthy donors.

## Histopathology and immunotyping

Mouse tissue samples were fixed in formalin (SF100-4; Fisher), embedded in paraffin (AC41677-0020; Fisher), sectioned into 5 μm tissue samples, and stained with hematoxylin and eosin (7211 & 7111, Fisher). For caspase-3 (AF835, 1:200; R&D Systems), PCNA (MS-106P, 1:200; Lab Vision Corp.), and B220 (14-0452-85, 1:100; eBioscience) detection, representative sections were deparaffinized and rehydrated in graded alcohols before subjected to antigen retrieval treatment with 10 mM sodium citrate buffer 10 min in a pressure cooker. Detection of antibody staining was carried out following standard procedures from the avidin-biotin immunoperoxidase methods. Diaminobenzidine (002014, Invitrogen) was used as the chromogen and hematoxylin as the nuclear counter stain. Quantitation of apoptosis was evaluated by counting the number of starry sky foci in three fields (40X) from seven representative animals of each genotype, as well as by counting the number of caspase-3 positive cells in three fields (40X) from five representative animals of each genotype.

To determine the cell surface markers of the lymphoma cells harvested from the animals, cells were resuspended in 10% FBS/PBS to reach a concentration of $10^7$ cells/ml. 20 μl of this cell suspension was stained with antibodies diluted in 10% FBS/PBS for 1 hr. Subsequently, cells were washed with 2% FBS/PBS and resuspended in 10% FBS/PBS for flow cytometry analysis. Antibodies used for FACS analyses include PE anti-mouse IgM (12-5790, 1:200; eBioscience), APC-Cy7 anti-mouse B220 (552094, 1:200; BD Pharmingen), APC-Cy7 anti-mouse CD4 (552051, BD Pharmingen, 1:200), PE anti-mouse CD8 (553032, 1:200; BD Pharmingen), PE anti-mouse CD25 (553866, 1:200; BD Pharmingen), and APC anti-mouse CD19 (115511, 1:100; Biolegend).

## Apoptosis assays and proliferation assays

Subconfluent MSCV- or *miR-92*-infected *R26^{MER/MER}* MEFs were induced and serum starved by incubating the cells with 100 nM of 4-hydroxytamoxifen (H6278; Sigma) in DMEM with 0.2% fetal bovine serum for 12–24 hr before harvesting the cells for apoptosis analyses using APC-Annexin V antibody (550475, 1:50; BD Pharmingen) and 7AAD staining solution (559925; BD Pharmingen). To evaluate the apoptotic effects of *miR-92* in our adoptive transfer model in vivo, we collected premalignant *Eμ-myc* B-cells from spleen or bone marrow of well-controlled *Eμ-myc/92* and *Eμ-myc/MSCV* mice at 5 weeks after adoptive transfer and measured the extent of apoptosis by FACS. Apoptosis in GFP-positive

B220-positive premalignant B-cells was measured using the Caspase Detection Kit (Calbiochem, Red-VAD-FMK) following the manufacturer's instructions. To quantitate cell proliferation, 10 µM of BrdU was used to label primary B-cells for 4 hr and MEFs for 30 min. The percentage of BrdU-positive cells was determined using the Flow BrdU kit (552598; BD Pharmingen).

## Real time PCR and western analyses

TaqMan MicroRNA Assays (Applied Biosystems) were used to measure the level of mature miRNAs, including *miR-17, 18, 19a, 20, 19b*, and *92* (4427975; ABI). mRNA level for *perp* (GACCCCAGAT GCTTGTTTTC, GGGTTATCGTGAAGCCTGAA), *noxa* (GGAGTGCACCGGACATAACT, TGAGCACACTC GTCCTTCAA), *puma* (GCGGCGGAGACAAGAAGA, AGTCCCATGAAGAGATTGTAC), p21 (ACGGT GGAACTTTGACTTCG, CAGGGCAGAGGAAGTACTGG), *bax* (GTTTCATCCAGGATCGAGCAG, CCCCAGTTGAAGTTGCCATC), *mdm2* (CTCTGGACTCGGAAGATTACAGCC, CCTGTCTGATA GACTGTCACCCG), *p53* (AACCGCCGACCTATCCTTAC, TCTTCTGTACGGCGGTCTCT), *ccnb1* (AAGGTGCCTGTGTGTGAACC, GTCAGCCCCATCATCTGCG), *ccnb2* (GCCAAGAGCCATGTGAC TATC, CAGAGCTGGTACTTTGGTGTTC), *cdc20* (AGACCACCCCTAGCAAACCT, GACCAGGCTTTC TGATGCTC), *cdc25b* (ATTCTCGTCTGAGCGTGGAC, GCTGTGGGAAGAACTCCTTG), *fbw7* (CGGCTCAGACTTGTCGATACT, CTTGATGTGCAACGGTTCAT), *gtse1* (GCTTTGCCTGTGAGAGGA AG, CACTCTGGGATCCCTTTTCA), *bid* (CTGCCTGTGCAAGCTTACTG, GTCTGGCAATGTT GTGGATG), *pten* (CACAATTCCCAGTCAGAGGCG, GCTGGCAGACCACAAACTGAG), *bim* (ACCA CTATCTCAGTGCAATGGCTTCC, CGGTAATCATTTGCAAACACCCTCCTTG), *cdk4* (TGGTACCGA GCTCCTGAAGT, GTCGGCTTCAGAGTTTCCAC), *c-myc* (GTGCTGCATGAGGAGACACCGCC, GCCCGACTCCGACCTCTTGGC), *Pirh2* (TGCAGTGCATCAACTGTGAA, CAAACAGGTGGCAAAT ACTGC), *Ppp2r5d* (CCGTGATGTTGTCACTGAGG, ACTCTGCTCCTGTGGGATTC), *Dyrk2* (CCAGCA ACGCTACCACTACA, AACAGCTGCTGAACCTGGAT), *Romo1* (ATTCGGAGTGAGACGTCGAG, TGACGAAGCCCATCTTCAC), *Pak2* (TTGGCTTTGATGCTGTTACG, CACTGCCTGAGGGTTCTTCT), *Trpc4ap* (CGCAAATGTCCTTCCTCTTC, GCCAGCATCAGGATTACCAG), and *Axin1* (AGGACG CTGAGAAGAACCAG, CTGCTTCCTCAACCCAGAAG) were determined using real time PCR analyses with SYBR (KK4605; Kapa Biosystems). *Actin* (GATCTGGCACCACACCTTCT, GGGGTGTTGAA GGTCTCAAA) was used as a normalization control in all our real time PCR analysis with SYBR. U6 snRNA assay (4427975; ABI) was used as a normalization control in all our TaqMan MicroRNA Assays (Applied Biosystems).

For western analyses, all samples were directly collected into Laemmli buffer. p53 (1C12; Cell Signaling), Arf (5-C3-1; Novus), and c-Myc (1472-1; Epitomics) antibodies were used at 1:1000 dilution. FLAG (M2; Sigma) and Tubulin (12G10) were used at 1:2500 dilution. HRP conjugated secondary antibodies (Santa Cruz Biotechnology, sc-2004 sc-2005 and sc-2006) were used at 1:5000.

## Microarray analyses

Three independent *R26^MER/MER* MEF lines were infected by MSCV vector alone or by MSCV vector encoding *miR-92*. These MEFs were induced and serum starved by incubating the cells with 100 nM of 4-hydroxytamoxifen (H6278; Sigma) in DMEM with 0.2% fetal bovine serum for 12 hr before harvesting the cells for RNA preparation. Total RNAs were prepared using Trizol (15596018; Invitrogen), and subjected to microarray analysis using Affymetrix chip Mouse 430_2. To identify differentially expressed genes that could be regulated by *miR-92*, we used gcRMA in the bioconductor package (*Wu et al., 2004*) and SAM (Significance Analysis of Microarrays) (*Tusher et al., 2001*) for statistical analysis of our microarray data. Gene expression signals were estimated from the probe signal values in the CEL files using statistical algorithm gcRMA. This data processing at the probe level includes background signal subtraction and quantile normalization to facilitate the comparison among microarrays. SAM was then used to identify the genes with significant expression level alterations between *miR-92* overexpressing MEFs and the control MEFs. The genes with at least 1.5-fold expression level change and FDR <1% were regarded as differentially expressed genes. Pathway analyses were performed on upregulated and downregulated genes using the KEGG database (*Dennis et al., 2003*).

## *Xenopus* embryo apoptosis assays

*Xenopus laevis* eggs were collected, fertilized, and embryos cultured by standard procedures. The *miR-19b* mimics were produced from the annealing products of 5'UGUGCAAAUCCAUGCAAAACUGA3' and 5'AGUUUUGCAGGUUUGCAUCCAUU3' (IDT).

The *miR-17* mimics were produced from the annealing products of 5'CAAAGUGCUUACAGUGCAG GUAGU3' and 5'UACUGCAGUGAAGGCACUUGUAG3'(IDT).

The *miR-18* mimics were produced from the annealing products of 5'UAAGGUGCAUCUAGUG CAGAUAG3' and 5'ACUGCCCUAAGUGCUCCUUCUG3'(IDT).

The *miR-19a* mimics were produced from the annealing products of 5'AGUUUUGCAUAGUUGC ACUA3' and 5'UGUGCAAAUCUAUGCAAAACUGA3'(IDT).

The *miR-20* mimics were produced from the annealing products of 5'UAAAGUGCUUAUAGUGC AGGUAG3' and 5'ACUGCAUAAUGAGCACUUAAAGU3'(IDT).

The *miR-92* mimics were produced from the annealing products of 5'UAUUGCACUUGUCCCGG CCUG3' and 5'AGGUUGGGAUUUGUCGCAAUGCU3'(IDT).

The annealing of miRNA mimics were performed by combining two complimentary RNA oligos at a stock concentration of 1 µg/µl, heating the oligos to 80°C for 1 min, and then cooling down to room temperature to allow duplexes to form. The same was done for generating the mutated *miR-19* mimics (*Mut-miR-19*), by annealing 5'UCAGGUAAUCCAUGCAAAACUGA3' and 5'AGUUUUGCAGGUUACCU UCGAUU3', and mutated *miR-92* mimics (*Mut-miR-92*) by annealing 5'UUAUCGACUUGUCCCGG3' and 5'GGUUGGGAUUGGUUCGA 3'.

*Xenopus* embryos were injected into both cells at the two-cell stage with 2 ng of each RNA (*Walker and Harland, 2009*). The pcDNA3-*myc-AGO2* vector, kindly provided by Dr Greg Hannon, was cut using Scal; and the synthetic *hAGO2* mRNAs were transcribed using mMessage mMachine T7 kit (Ambion). When indicated, a total of 0.5 ng *hAGO2* mRNA (*Liu et al., 2004*) was injected into two-cell stage embryos either alone or with 2 ng of each miRNA (*Lund et al., 2011*). The embryos were then treated with hydroxyurea (H8627; Sigma) at a final concentration of 5 mM from stage 3 until stage 10. Apoptotic embryos were scored as those containing any apoptotic cells based on morphological changes.

## Luciferase assays

A luciferase reporter fused with the *fbw7* 3'UTR was kindly provided by Dr Hans-Guido Wendel (*Mavrakis et al., 2011*). In this psiCHECK-2 based reporter, the *fbw7* 3'UTR was cloned downstream of the *Renilla* luciferase reporter, and a separate *firefly* luciferase cassette was used as a transfection control. Because the two predicted *miR-92* binding sites are close to each end of the 3'UTR, we mutated the *miR-92* binding sites by PCR using the following primers:

3'UTR-Fbw7-Mut-Xho1-F (GATCTCGAGCAAGACGACTCTCTAAATCCAACTATTCTTT) and 3'UTR-Fbw7-mut-Not1-R (ATGCGGCCGCAACACATTTAGTTATAAGAAAATAAAATTT). The PCR fragment was subsequently cloned into the XhoI and Not1 sites of the psiCHECK-2 vector. The reporter construct, together with 50 nM *miR-92* mimics, was transfected into *Dicer*-deficient Hct116 cells (*Cummins et al., 2006*), with transfection of *miR-17* or *miR-18* as negative controls. Luciferase activity of each construct was determined by dual luciferase assay (E19100; Promega) 48 hr post-transfection following the manufacturer's instructions. The *miR-17* mimics were produced by annealing 5'CAAAGUGCUUAC AGUGCAGGUAGU3' and 5'UACUGCAGUGAAGGCACUUGUAG3'. The *miR-18* mimics were produced by annealing 5'UAAGGUGCAUCUAGUGCAGAUAG3' and 5'ACUGCCCUAAGUGCUCC UUCUG3'.

The *miR-92* mimics were produced by annealing 5'UAUUGCACUUGUCCCGGCCUG3' and 5'AGGUUGGGAUUUGUCGCAAUGCU3'.

## Fbw7α immunoprecipitation and western analyses

Because Fbxw7-substrate degradation was regulated in a cell-cycle-dependent manner, we used serum starvation synchronized MEFs to study Fbw7 regulation by *miR-92* during cell cycle progression. MEFs were made quiescent by serum starvation; then Fbw7 expression was examined following release into serum. Cells were lysed in NP-40 buffer supplemented with protease inhibitors. Lysates were normalized and immunoprecipitated with polyclonal anti-Fbw7 antibody kindly provided by Dr Bruce Clurman (*Grim et al., 2008*), followed by immunoblotting with polyclonal anti-Fbw7 antibody (A301-720A; Bethyl Laboratories). Wild-type and *FBW7*−/− Hct116 cells were used, respectively, as positive and negative controls.

The construction of the pFLAG-Fbw7α-3'UTR plasmid was previously described (*Xu et al., 2010*). The construct was transfected into the *Dicer*-deficient Hct116 cells together with 50 nM of *miR-92* mimics or siRNA against GFP as indicated. Anti-FLAG (M2; Sigma) antibody was used to detect the FLAG-Fbw7α by western blot 48 hr after transfection.

## Cyclin E-dependent kinase assays

Cyclin E-CDK complexes were immunoprecipitated from MSCV or *miR-92* infected *Rosa26*[MER/MER] MEFs extracts using affinity-purified polyclonal antibody, provided by Dr Bruce Clurman (*Minella et al., 2008*). Cyclin E immunoprecipitates were then incubated with purified histone subunit H1 (Sigma) and (gamma-$^{32}$P)ATP to measure cyclin E-dependent kinase activity. The anti-Grb2 monoclonal (BD Biosciences) antibody was used a normalization control.

## Acknowledgements

We thank members of the He Lab for their help and input. Particularly, we thank C Fulco, I Jiang, Y Chen, R Song, E Ho, J Cisson, YJ Choi and C Lin for technical assistance and stimulating discussions. We also thank H Nolla and A Valeros for advice on our FACS analysis, thank J Choi for microarray analyses, and P Margolis for proofreading our manuscript. We thank SW Lowe, J Mendell, B Clurman, M Schlissel, M Junttila, L Soucek, HG Wendel, A Ventura, GJ Hannon, DS Sandeep, T Rabbitts, B Vogelstein, J Mao and M Burger for sharing reagents and helpful discussions. We are particularly grateful for B Olive and B Colpo for their support during this study. Finally, we would like to dedicate this work to the memory of Gisele Cocher, whom we lost during the preparation of this manuscript. Her unconditional love and kindness shape who we are; her courage and support will always be with us. LH is a Searle Scholar supported by the Kinship Foundation.

## Additional information

### Funding

| Funder | Grant reference number | Author |
| --- | --- | --- |
| American Cancer Society | 123339-RSG-12-265-01-RMC | Lin He |
| National Cancer Institute | R00 CA126186 | Lin He |
| Tobacco-Related Disease Research Program | 21RT-0133 | Lin He |
| The Leukemia and Lymphoma Society | LLS, 3423-13 | Virginie Olive |
| National Institite of Health | F31 CA165825-02 | Erich Sabio |
| National Cancer Institute | R01 CA139067 | Lin He |
| National Cancer Institute | 1R21CA175560-01 | Lin He |
| National Institute of Health | R01HL098608 | Alex C Minella |
| National Heart, Lung and Blood Institute | R01HL098608 | Alex C Minella |
| US Department of Defense | W81XWH-12-1-0272 | Andrei Goga |
| National Institutes of Health | 5R01CA170447 | Andrei Goga |
| The Leukemia and Lymphoma Society | LLS, 1531 | Andrei Goga |

The funders had no role in study design, data collection and interpretation, or the decision to submit the work for publication.

### Author contributions

VO, ES, ACM, LH, Conception and design, Acquisition of data, Analysis and interpretation of data, Drafting or revising the article; MJB, Acquisition of data, Analysis and interpretation of data, Drafting or revising the article; CSDJ, Conception and design, Acquisition of data; AB, Acquisition of data, Analysis and interpretation of data; JCM, NMS, TPS, GIE, YW, Conception and design, Acquisition of data, Analysis and interpretation of data; SKG, AYZ, AB, MF, MAL, AG, Acquisition of data, Analysis and interpretation of data, Contributed unpublished essential data or reagents; ZX, Conception and design, Analysis and interpretation of data

### Ethics

Animal experimentation: Our experimentation is conducted to the highest ethical standards, and we follow the guidelines established by the University of California, Berkeley's Animal Care and Use

Committee (ACUC). The animal protocol detailing the experimental procedures with laboratory mice was carefully reviewed and approved by Animal Care and Use Committee (ACUC) at the University of California at Berkeley. Our Animal Use protocol number is R316-0613BR.

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
