## [Decision Letter]

Thank you for sending your work entitled “A component of the *mir-17-92* polycistronic oncomir promotes oncogene-dependent apoptosis” for consideration at *eLife*. Your article has been favorably evaluated by a Senior editor and 3 reviewers, one of whom, Chi Van Dang, is a member of our Board of Reviewing Editors.

The Reviewing editor and the other reviewers discussed their comments before we reached this decision, and the Reviewing editor has assembled the following comments to help you prepare a revised submission.

The manuscript by Olive et al. describes a very intriguing finding: while as a whole the *miR-17∼92* cluster accelerates Myc-driven lymphomagenesis, its *miR-92* component acts as a built-in damper, which induces apoptosis. Furthermore, deleting this component results in earlier-onset lymphomas. This central discovery was made using a model developed by Dr. He and her collaborators (Nature 2005; G&D 2009), wherein premalignant hematopoietic progenitors from *Eμ-myc* mice are transduced with various miR-encoding retroviruses and used to reconstitute irradiated recipients. The key data presented in Figures 1, 2 and 3 are generally crisp, compelling, and easy to interpret. However several issues remain to be resolved.

*Specificity of* miR-92 *activity:*

1) The authors documented that *miR-92* targets Fbw7 and thereby enhances MYC protein levels. While Figure 5 shows that *miR-92* can repress a Fbw7 3'UTR luciferase reporter construct or Fbw7 cDNA expression construct, specificity of *miR-92* should be established via mutating the predicted *miR-92* binding site(s) within the 3'UTR and determine whether they are required for this repression.

2) Further experiments are needed to show whether the *miR-92*-Fbw7-Myc axis is fully responsible for *miR-92*'s pro-apoptotic effects. In Figure 5, it is shown that Fbw7 shRNA partially recapitulates the effect of *miR-92* expression on Myc-mediated apoptosis in MEFs. The levels of Myc protein should be shown in this experiment. If Fbw7 knockdown fully recapitulates the *miR-92*-induced Myc levels yet does not fully recapitulate the degree of *miR-92*-induced apoptosis, it suggests that *miR-92* engages additional mechanisms to induce apoptosis. To further investigate this issue, the authors should express ectopic Fbw7 (which does not have the miR seed sequence) at physiologic levels and see if this rescues the apoptotic phenotype of *miR-92*. These experiments should establish whether upregulation of Myc by *miR-92* is the entire story or whether additional pro-apoptotic mechanisms exist. The authors are not required to identify such additional mechanisms in the current paper but it is important to know whether they exist.

3) To further substantiate their model, the authors should measure Fbw7 and Myc protein levels in *Eμ-myc* lymphoma cells expressing wild-type *miR-17-92* versus those expressing *miR-17-92Δ92*. If *miR-92*-mediated Fbw7 repression/Myc induction cannot be demonstrated in this setting, the proposed mechanism, while elegant, could be completely irrelevant.

4) Along the same lines, what is the evidence that the effects of *miR-92* on Myc levels are Fbw7-mediated? Perhaps Fbw7-null HCT116 cells could be used to establish causality.

5) The authors make a claim that that *miR-92* is processed less efficiently in murine and human lymphomas than *miR-19*, the main oncogenic component of the cluster. This is an important claim, however, some of the quantifications are difficult to understand. For example, in panel 7D, *miR-19* and *-92* levels in Burkitt's cell lines are normalized separately for those found in “normal PB-cells”. Assuming that PB stands for “peripheral blood”, this does not appear to be a relevant control, since a circulating lymphocyte is not a cell of origin for Burkitt's – or other human lymphomas, for that matter. A direct comparison between *miR-19* and *miR-92* levels would be more helpful. According to a recent profiling paper [doi:10.1038/bcj.2012.1], *miR-19b* and *miR-92* are overexpressed at comparable levels in Burkitt's samples.

*Conceptual framework*:

1) Although the authors focused on this ‘oncoMir’ cluster and studied its oncogenic properties, it would be terrific for the authors to discuss the potential physiological importance of this cluster with regard to its evolution as presented in the manuscript. In particular, it would be safe to assume that this cluster evolved to regulate cell growth and proliferation downstream or independent of MYC. Hence, the different miRs in the cluster might be subject to regulation via microRNA processing in addition to the expression of the cluster mRNA precursor. In this regard, are the relative levels of *miR-92* to other miRs in the cluster differentially affected by cellular stresses that lead to apoptosis (serum or growth factor deprivation, nutrient deprivation?)? Some discussion on this aspect of *miR-17-92* function could be very useful for the field.

2) In the Discussion, the authors describe *miR-92* as conferring negative feedback on the oncogenic activity of *miR-17-92*. Given that *miR-17-92* is transcriptionally activated by Myc and Myc dosage is positively regulated by *miR-92*, a positive feedback loop is also established. This concept should be discussed.

*Influence of* miR-92 *on the miR cluster*:

1) The expression of the miRNAs derived from the various MSCV constructs (*miR-17-92*; *miR-17-92Δ92*; *miR-17-92Mut92*) is tested by transducing 3T3 cells with these retroviruses (Figure 1). However, the conclusions of the paper rest heavily upon the assumption that mutating *miR-92* does not affect the expression of other miRNAs in the cluster in B cells (where the oncogenic activity is examined most extensively). Therefore it is important to examine the miRNA levels in *Eμ-Myc* lymphoma cells or primary B cells infected with the various viruses to confirm their findings in 3T3 cells.

---

## [Author Response]

*Specificity of* miR-92 *activity*:

*1) The authors documented that* miR-92 *targets Fbw7 and thereby enhances MYC protein levels. While*
Figure 5
*shows that* miR-92 *can repress a Fbw7 3'UTR luciferase reporter construct or Fbw7 cDNA expression construct, specificity of* miR-92 *should be established via mutating the predicted* miR-92 *binding site(s) within the 3'UTR and determine whether they are required for this repression*.

We thank the reviewers for this comment. We have constructed luciferase reporters that carry either a wild type *fbw7* 3’UTR, or a mutated *fbw7* 3’UTR with defective *miR-92* binding sites. Using these reporters, we clearly demonstrated that *miR-92* overexpression could downregulate the expression of the luciferase reporter carrying the wild type *fbw7* 3’UTR, but not the luciferase reporter with the mutated *fbw7* 3’UTR. This result, shown in Figure 5, demonstrates that Fbw7 is specifically repressed by *miR-92*, and that the *miR-92* binding is required for its repression on Fbw7.

*2) Further experiments are needed to show whether the* miR-92*-Fbw7-Myc axis is fully responsible for* miR-92*'s pro-apoptotic effects. In*
Figure 5*, it is shown that Fbw7 shRNA partially recapitulates the effect of* miR-92 *expression on Myc-mediated apoptosis in MEFs. The levels of Myc protein should be shown in this experiment. If Fbw7 knockdown fully recapitulates the* miR-92*-induced Myc levels yet does not fully recapitulate the degree of* miR-92*-induced apoptosis, it suggests that* miR-92 *engages additional mechanisms to induce apoptosis. To further investigate this issue, the authors should express ectopic Fbw7 (which does not have the miR seed sequence) at physiologic levels and see if this rescues the apoptotic phenotype of* miR-92*. These experiments should establish whether upregulation of Myc by* miR-92 *is the entire story or whether additional pro-apoptotic mechanisms exist. The authors are not required to identify such additional mechanisms in the current paper but it is important to know whether they exist*.

To investigate if the *miR-92*-Fbw7-Myc axis is fully responsible for *miR-92*'s pro-apoptotic effects in vitro, we compared the effect of *miR-92* overexpression and *fbw7* knockdown on c-Myc protein level in *R26*^*MER/MER*^ mouse embryonic fibroblasts (MEFs) (Figure 5—figure supplement 1). In this experiment, *fbw7* knockdown largely recapitulated the extent of c-Myc upregulation by *miR-92*. This is consistent with our observation that the repression of *fbw7* by *miR-92* is essential for *miR-92* to upregulate c-Myc (Figure 5—figure supplement 1, also see our response to #4). Since *fbw7* knockdown only partially phenocopies *miR-92* in promoting c-Myc induced apoptosis, one possible scenario is that *miR-92* engages additional mechanisms to promote c-Myc apoptosis. Nevertheless, the *miR-92*-Fbw7-Myc axis does constitute a major mechanism to mediate the pro-apoptotic effects of *miR-92*. To examine the importance of *fbw7* in mediating the apoptotic effects by *miR-92*, we expressed *fbw7* in *R26*^*MER/MER*^ mouse embryonic fibroblasts (MEFs) with and without *miR-92* overexpression. In this experiment, the *fbw7* cDNA introduced did not contain its 3’UTR, thus was not regulated by *miR-92*. Although *miR-92* overexpression in *R26*^*MER/MER*^ MEFs invariably enhanced c-Myc induced apoptotic response upon MycERT(3) activation, expression of *fbw7* abolished this apoptotic effect of *miR-92* (Figure 5). Thus, the ability of *miR-92* to increase c-Myc protein level through *fbw7* repression constitutes the major mechanism underlying its pro-apoptotic effects.

*3) To further substantiate their model, the authors should measure Fbw7 and Myc protein levels in* Eμ-myc *lymphoma cells expressing wild-type* miR-17-92 *versus those expressing* miR-17-92Δ92*. If* miR-92*-mediated Fbw7 repression/Myc induction cannot be demonstrated in this setting, the proposed mechanism, while elegant, could be completely irrelevant*.

We thank the reviewers for this insightful comment. The experiment proposed here, if performed successfully, would strongly support our hypothesis. However, we have encountered technical limitations in detecting the endogenous Fbw7 protein in our tumor lysates. In our experience, we have not found any Fbw7 antibodies that can reliably detect endogenous Fbw7 proteins by simple immunoblotting. We have tested several commercial antibodies for detection of endogenous Fbw7, including those sold by Abcam, Sigma, and Invitrogen, and we are unable to detect endogenous Fbw7 cleanly, using proper controls (Fbw7-null HCT116 cell lysate). In lysates from cultured MEFs, which we can expand greatly, we use an immunoprecipitation-western blot method that does detect endogenous Fbw7 (Figure 5), as detailed in our Methods section. The limitation of this approach is that one needs a large amount of cell pellet for this experiment. As an alternative, we performed *fbw7* QPCR analyses, using *Eμ-myc/17-92*, *Eμ-myc/17-19b*, and *Eμ-myc/MSCV* lymphoma cells. Consistent with our hypothesis, Eμ-myc/17-92 B-lymphoma cells exhibited significantly decreased levels of *fbw7* mRNA, when compared to those in *Eμ-myc/17-19b* or *Eμ-myc/MSCV* lymphoma cells (Figure 5—figure supplement 1).

We also measured c-Myc protein levels in several lines of *Eμ-myc/17-92*, *Eμ-myc/17-19b*, and *Eμ-myc/MSCV* lymphoma cells, to determine if there is a correlation between *miR-92* overexpression and increased c-Myc dosage. However, we observed no differences in the c-Myc protein levels among these terminal tumor cells (data not shown). Previous studies have demonstrated that the terminal *E-myc* tumors, which are defective for c-Myc-induced apoptosis, clearly favor a high dosage of c-Myc to promote and maintain oncogenesis. In addition to the *miR-92*-Fbw7 axis that regulates c-Myc dosage, a *miR-92* and*fbw7* independent mechanism can also enhance c-Myc dosage in the transformed *Eμ-myc* lymphoma cells. Thus, comparing the c-Myc level in the terminal *Eμ-myc/17-92*, *Eμ-myc/17-19b*, and *Eμ-myc/MSCV* lymphoma cells is unlikely to reveal the importance of c-Myc regulation by the *miR-92*-Fbw7 axis, because this regulation plays an essential role in the early stages of lymphoma development (Figure 3, Figure 6).

*4) Along the same lines, what is the evidence that the effects of* miR-92 *on Myc levels are Fbw7-mediated? Perhaps Fbw7-null HCT116 cells could be used to establish causality*.

In the revised manuscript, we have clearly demonstrated that the overexpression of *miR-92* increases c-MYC protein levels in a *FBW7*-dependent manner. The effect of *miR-92*to upregulate c-MYC protein level was observed in wild type Hct116 cells, but was largely absent in *FBW7*^-/-^ Hct116 cells (Figure 5—figure supplement 1). These results argue that the repression of *FBW7* by *miR-92* is essential for *miR-92* to upregulate the protein level of c-MYC.

*5) The authors make a claim that that* miR-92 *is processed less efficiently in murine and human lymphomas than* miR-19*, the main oncogenic component of the cluster. This is an important claim, however, some of the quantifications are difficult to understand. For example, in panel 7D,* miR-19 *and* -92 *levels in Burkitt's cell lines are normalized separately for those found in “normal PB-cells”. Assuming that PB stands for “peripheral blood”, this does not appear to be a relevant control, since a circulating lymphocyte is not a cell of origin for Burkitt's – or other human lymphomas, for that matter. A direct comparison between* miR-19 *and* miR-92 *levels would be more helpful. According to a recent profiling paper [**doi:10.1038/bcj.2012.1**],* miR-19b *and* miR-92 *are overexpressed at comparable levels in Burkitt's samples*.

We thank the reviewers for the constructive comments. We have realized that our wording in the previous manuscript has caused confusion. What is clear from our studies is that the ratio of *miR-19* to *miR-92* is greater in B-lymphomas than in normal B-cells. In other words, when normalized to the respective miRNA levels in normal B-cells, mature *miR-19* exhibited a greater increase in premalignant and malignant *Eμ-myc* B-cells than mature *miR-92* (Figure 7). Since *miR-19*and *miR-92* are coregulated transcriptionally, we speculate, but do not claim, that a differential miRNA biogenesis and/or turn over could explain the differential increase of these two miRNAs. Given their functional antagonism, the ratio between *miR-19* and *miR-92* is the key determinate for the oncogenic activity of *mir-17-92* in the context of the *Eμ-myc* B-lymphoma model. What we showed here strongly supported an altered *miR-19*:*miR-92* ratio in premalignant and malignant *Eμ-myc* B-cells, which favored a greater *miR-19* increase to drive oncogenesis.

Per the reviewers’ request, we directly compared the *miR-19* and *miR-92* levels using miRNA Taqman asssays. Our data suggest a ∼2-5 fold increase in the absolute level of *miR-19b* than *miR-92* in transformed B-cells, both in mouse and in human (data not shown). However, we must point out the intrinsic caveats associated with absolute quantitation of different mature miRNAs. Currently, two methods are most popular for the absolute quantitation of mature miRNAs miRNA Taqman assays or high-throughput sequencing (HTS). However, both methods have technical caveats that prevent an accurate quantitation. For the Taqman miRNA assays, the different RT efficiency for different mature miRNAs can introduce systematic bias in quantitation and preclude an accurate quantitation of different mature miRNAs. For the HTS approach, different mature miRNAs have different cloning efficiency due to RNA-ligase-dependent bias (Hafner et al., RNA 2011). Given the intrinsic technical limitations to accurately compare copy numbers of different mature miRNAs, we think it is the most appropriate to leave this out for our manuscript. We included a discussion about this issue in the revised manuscript.

We also clarified the legend of our Figure 7 to indicate the use of normal B-cells from periphery blood as a control for our Burkitt’s lymphoma cell lines. We admit that using B-cells from peripheral blood to control for human Burkitt’s lymphoma cell lines is less than ideal. However, such comparison has been used routinely for many published studies due to the difficulty to acquire human GC B-cell RNA as a proper control. We have included a statement in our revised manuscript to discuss this caveat for our comparison.

Conceptual framework:

*1) Although the authors focused on this ‘oncoMir’ cluster and studied its oncogenic properties, it would be terrific for the authors to discuss the potential physiological importance of this cluster with regard to its evolution as presented in the manuscript. In particular, it would be safe to assume that this cluster evolved to regulate cell growth and proliferation downstream or independent of MYC. Hence, the different miRs in the cluster might be subject to regulation via microRNA processing in addition to the expression of the cluster mRNA precursor. In this regard, are the relative levels of* miR-92 *to other miRs in the cluster differentially affected by cellular stresses that lead to apoptosis (serum or growth factor deprivation, nutrient deprivation?)? Some discussion on this aspect of* miR-17-92 *function could be very useful for the field*.

We thank the reviewers for the constructive comment. We have included a brief discussion on the functional significance of the *mir-17-92* polycistronic structure in its physiological functions.

*2) In the Discussion, the authors describe* miR-92 *as conferring negative feedback on the oncogenic activity of* miR-17-92*. Given that* miR-17-92 *is transcriptionally activated by Myc and Myc dosage is positively regulated by* miR-92*, a positive feedback loop is also established. This concept should be discussed*.

We thank the reviewers for the insightful comment. In the revised manuscript, we have included a discussion on the positive feedback loop between *mir-17-92* and c-Myc.

Influence of *miR-92* on the miR cluster:

*1) The expression of the miRNAs derived from the various MSCV constructs (*miR-17-92*;* miR-17-92Δ92*;* miR-17-92Mut92*) is tested by transducing 3T3 cells with these retroviruses (*Figure 1*). However, the conclusions of the paper rest heavily upon the assumption that mutating* miR-92 *does not affect the expression of other miRNAs in the cluster in B cells (where the oncogenic activity is examined most extensively). Therefore it is important to examine the miRNA levels in* Eμ-Myc *lymphoma cells or primary B cells infected with the various viruses to confirm their findings in 3T3 cells*.

We have examined the expression of all *mir-17-92* components in the *Eμ-myc* B-lymphoma cells that overexpress *mir-17-92*, *mir-17-92Δ92*, or *mir-17-92Mut92*. Consistent with our results in the 3T3 cells (Figure 1—figure supplement 1), mutation or deletion of *miR-92* specifically disrupted the *miR-92* expression in B-cell, without affecting the expression of the remaining *mir-17-92* components (Figure 1).